# Ecotype-specific blockage of tasiARF production by two different RNA viruses in *Arabidopsis*

**Péter Gyula**[1][ⓥ]*, **Tamás Tóth**[1,2][ⓥ], **Teréz Gorcsa**[1], **Tünde Nyikó**[1], **Anita Sós-Hegedűs**[1], **György Szittya**[1]*

**1** Department of Plant Biotechnology, Institute of Genetics and Biotechnology, Hungarian University of Agriculture and Life Sciences, Gödöllő, Hungary, **2** Doctoral School of Biology, Institute of Biology, ELTE Eötvös Loránd University, Budapest, Hungary

ⓥ These authors contributed equally to this work.
* gyula.peter@uni-mate.hu (PG); szittya.gyorgy@uni-mate.hu (GS)

**Data Availability Statement:** The raw sequencing data were deposited in the NCBI SRA database (BioProject: PRJNA788379) under the identifiers SRR17227515–SRR17227578. All the other

## Abstract

*Arabidopsis thaliana* is one of the most studied model organisms of plant biology with hundreds of geographical variants called ecotypes. One might expect that this enormous genetic variety could result in differential response to pathogens. Indeed, we observed previously that the Bur ecotype develops much more severe symptoms (upward curling leaves and wavy leaf margins) upon infection with two positive-strand RNA viruses of different families (turnip vein-clearing virus, TVCV, and turnip mosaic virus, TuMV). To find the genes potentially responsible for the ecotype-specific response, we performed a differential expression analysis of the mRNA and sRNA pools of TVCV and TuMV-infected Bur and Col plants along with the corresponding mock controls. We focused on the genes and sRNAs that showed an induced or reduced expression selectively in the Bur virus samples in both virus series. We found that the two ecotypes respond to the viral infection differently, yet both viruses selectively block the production of the *TAS3*-derived small RNA specimen called tasiARF only in the virus-infected Bur plants. The tasiARF normally forms a gradient through the adaxial and abaxial parts of the leaf (being more abundant in the adaxial part) and post-transcriptionally regulates ARF4, a major leaf polarity determinant in plants. The lack of tasiARF-mediated silencing could lead to an ectopically expressed ARF4 in the adaxial part of the leaf where the misregulation of auxin-dependent signaling would result in an irregular growth of the leaf blade manifesting as upward curling leaf and wavy leaf margin. QTL mapping using Recombinant Inbred Lines (RILs) suggests that the observed symptoms are the result of a multigenic interaction that allows the symptoms to develop only in the Bur ecotype. The particular nature of genetic differences leading to the ecotype-specific symptoms remains obscure and needs further study.

## Introduction

Virus-infected plants may develop various symptoms on all or some of their parts. The symptoms may either be nonspecific or quite characteristic of the particular virus. Common

relevant data are within the article and its
Supporting Information files.

**Funding:** This work was funded by the Hungarian
Government organization NRDI (National
Research, Development, and Innovation Office:
https://nkfih.gov.hu/about-the-office) through the
grants K-119701, K-129171, K-134974 awarded to
GS, FK-137811 awarded to PG, and PD-129119
awarded to TN. The funders had no role in study
design, data collection and analysis, decision to
publish, or preparation of the manuscript.

**Competing interests:** The authors have declared
that no competing interests exist.

symptoms are dwarfing, stunting, various leaf deformations, chlorotic mosaics, vein clearing, etc. These effects may either be severe or hardly detectable. Many viruses may infect certain hosts without causing visible symptoms. Such viruses are called latent viruses and the hosts are called symptomless carriers. The development of symptoms usually depends on environmental conditions, like temperature and light. Finally, plants may show acute or severe symptoms soon after inoculation that may lead to the death of the host. If the host survives the initial shock, the symptoms tend to become milder in the subsequently developing part of the plant, leading to partial or total recovery. On the other hand, symptoms may progressively increase in severity and may result in a gradual decline of the plant [1]. Virus infection and symptom development is a multistep process. First, the virus must invade the plant cell either by mechanical means or by a vector. After uncoating, inside the host cytoplasm, the viral RNA is translated by the host translational machinery and the virally encoded proteins become active. Movement proteins help the viral nucleic acid move into another cell through plasmodesmata [2]. At the site of entry, plants carrying appropriate R genes may develop a hypersensitive response that involves the overproduction of reactive oxygen species and programmed cell death which prevents further spreading of the virus into the neighboring cells. This manifests in local tissue lesions (local symptoms). If the virus manages to overcome this barrier and reaches the veins, it can spread into other tissues or organs causing systemic symptoms, i.e., leaf deformations, mosaics, yellowing, ringspots, mottling, stunting, and necrosis.

To restrict viral accumulation, plants and other eukaryotes utilize a highly adaptive, sequence-specific mechanism called RNA silencing [3]. Upon virus infection, the viral nucleic acid (i.e., positive single-stranded RNA) gets into the host cytoplasm, where it functions as a genuine mRNA. Double-stranded hairpin-like structures of this RNA molecule or the double-stranded RNA (dsRNA) intermediate product of viral replication trigger antiviral silencing [4]. The dsRNA is randomly cut into 21–22 bp-long fragments called virus-derived short interfering RNAs (vsiRNAs) by specific Dicer-like (DCL) endonucleases [5]. One of the strands of these fragments called the guide strand is then loaded into an RNA-induced silencing complex (RISC). With the help of the guide RNA, the RISC binds to the complementary single-stranded viral RNA molecules and either catalytically cleaves them or blocks their translation. In the first case, the cleavage products are completely degraded by exonucleases purging the viral RNA from the cytoplasm. For efficient virus infection, viruses should be able to evade the silencing machinery. In the evolutionary arms race, viruses developed a plethora of different viral silencing suppressor (VSR) molecules that can interfere with various steps of the silencing pathway, including small RNA production, processing, binding, stability, and activity of the RISC [6–8]. As a side effect, VSRs can interfere with endogenous sRNA-regulated processes that result in a corrupted development of the plant.

One major class of regulatory sRNAs is the micro RNAs (miRNAs) which are usually 20–22-nt-long sequences produced by a DCL1-mediated dicing of a hairpin-like, double-stranded precursor with imperfect self-complementarity. Different AGOs preferentially bind miRNAs with a specific 5' nucleotide [9, 10] and length while the loading efficiency is controlled by the structure of the double-stranded precursor [11]. Viruses often modify the miRNA landscape of the plant which affects symptom development [12–14].

Many aspects of plant development where a pattern formation is required are regulated by cell-to-cell mobile miRNAs or siRNAs [15–17]. This cell-to-cell movement results in a sharp gradient of the sRNA within a tissue, for example, between the adaxial and abaxial parts of the leaf, establishing patterns of cell specification by creating boundaries that limit the activity of target genes [17, 18]. A conserved regulatory module among the land plants that plays a central role in abaxial-adaxial leaf patterning, juvenile-to-adult phase transition, floral organ development, and lateral root development is the miR390/AGO7/TAS3/tasiARF/ARF3/ARF4 module

[19–28]. The tasiARF (*trans*-acting ARF-targeting small interfering RNA) is a special kind of small interfering RNA (siRNA) that is produced from the *TAS3* non-coding RNA (ncRNA) precursor. During the production of tasiARF, the *TAS3* transcript is first cleaved by a miR390-loaded AGO7 [29], then, a double-stranded intermediate product is produced by the RNA-dependent RNA polymerase RDR6/SGS3/DRB4 complex. This dsRNA is then diced in a phased manner, mostly by DCL4, producing a set of 21-nt-long siRNAs. From this siRNA population, only a few sequences are conserved evolutionary. The conserved tasiARF is loaded into AGO1 and post-transcriptionally regulates the transcripts of *ARF3* and *ARF4* transcription factor genes, the protein product of which negatively regulates auxin-dependent gene expression [20, 23, 30]. The tasiARF is cell-to-cell mobile and forms a gradient across the leaf blade being more abundant on the adaxial side [18, 31]. This results in the confinement of *ARF4* in the abaxial side of the leaf and contributes to the determination of abaxial cell identity.

Another sRNA-mediated signaling pathway that is conserved among land plants is the miRNA-regulation of the large family of *NUCLEOTIDE-BINDING LEUCINE-RICH REPEAT CONTAINING RECEPTOR* genes (NB-LRRs) which are important regulators of innate immunity. They recognize specific pathogen effectors and trigger resistance responses [32, 33]. The members of this gene family are targeted by various, unrelated, lineage-specific 22-nt-long miRNAs, some of which are more widespread (i.e. the miR482/2118 superfamily) [34], while others can be found only in certain plant groups [35]. These genes are constantly repressed by the above-mentioned miRNAs but relieved from repression when the level of the regulatory miRNA is lowered upon pathogen infection. Interestingly, in the presence of a symbiont, the miRNA level is changed in the opposite direction leading to an even lower level of NB-LRR genes and enhanced nodulation in *Medicago truncatula* [36]. The miRNA cleavage of the transcripts can initiate the production of secondary phasiRNAs: dsRNA intermediates are produced by the RDR6/SGS3/DRB4 complex and then diced by DCL4/DCL2, similarly to the case of *TAS3*. These secondary siRNAs can also target related genes amplifying the immune response [37].

There are other tasiRNA- or phasiRNA-producing loci in plants that are less conserved as the *TAS3* regulatory module [30]. In *Arabidopsis thaliana*, *TAS1a,b,c*, and *TAS2* noncoding RNA are cleaved by miR173-loaded AGO1 [38], while TAS4 is cleaved by miR828 [39] and secondary siRNAs are produced as described above. Some siRNAs emerging from these loci target host genes: *TAS1*-derived siRNAs can target *HEAT-INDUCED TAS1 TARGET 1* and *2* (*HTT1/2*) [40], *TAS2*-derived siRNAs can target several *PENTATRICOPEPTIDE REPEAT-CONTAINING PROTEIN* (*PPR*) genes [30], while a *TAS4*-derived siRNA can target a set of MYB transcription factor genes [39]. Other secondary phasiRNA-producing loci are mostly protein coding genes and include the miR393-initialized auxin receptor genes *TRANSPORT INHIBITOR RESPONSE 1* (*TIR1*) and *AUXIN SIGNALING F-BOX 1–3* (*AFB1–3*) that regulate auxin signaling [41].

The presence of the virus also triggers the synthesis of a different class of siRNAs (virus-activated siRNAs, vasiRNAs) that regulate other host genes and modulate the antiviral response of the plant [42], for example, by downregulating genes coding for different functions of the photosynthetic apparatus. Lowering the energy production rate within the virus-infected plant cell would restrict the replication of the virus which requires a lot of energy [43]. This conserved regulatory action of vasiRNAs against the host metabolism could explain the observation that almost all virus-infected plant show some degree of retarded growth. The production of vasiRNAs was shown to be dependent on RNA-DEPENDENT RNA POLYMERASE 1 (RDR1) and DICER-LIKE 4 (DCL4) [42] making them unique among other endogenous siRNAs like trans-acting siRNAs (tasiRNAs) or phased siRNAs (phasiRNAs) which depend on

the action of RDR6 and DCL4/DCL2. Unlike tasiRNAs, vasiRNAs do not require a phase initiating miRNA- or siRNA-mediated cleavage and, therefore, they are not produced in a phased manner. Pitzalis et al [44] proposed a mechanism for the triggering of vasiRNA production: virus infection could induce a disruption or overload of RNA decay pathways, thereby leading to the accumulation of aberrant transcripts that are prone to dsRNA formation and processing into siRNAs. Most of the vasiRNAs are 21-nt-long [42] but in some cases, for example, during infection by the pararetrovirus CaMV, a significant amount of 22-nt-long vasiRNAs are also produced in three different species of the *Brassicaceae* family, probably due to the activity of DCL2 [43]. The vasiRNAs can target the cleavage of the same mRNA that they were derived from or other genes *in trans*, similarly to tasiRNAs.

Viral infections often lead to severe diseases in plants that result in significant economic loss. Tobamoviruses are among the most studied plant viruses because they are widespread and infect economically important plants, including the large family of *Solanaceae* (i.e., potato, tomato, pepper, eggplant, etc.) and *Brassicaceae* (i.e., cruciferous vegetables like cabbage, turnip, mustard, rapeseed, etc.). The model plant of *Brassicaceae* is *Arabidopsis thaliana* (thale cress) which has several geographical varieties called ecotypes. We previously characterized a new isolate of *Turnip vein-clearing virus* (TVCV-ApH) [45], a crucifer-infecting tobamovirus [46, 47], that caused severe leaf deformation (upward curling leaves) on the Bur ecotype of *Arabidopsis thaliana*, but not on Col ecotype. The 6.3 kb genome of TVCV-ApH contains four ORFs. ORF1 codes for the 125 kDa small replicase subunit, while the suppression of an amber termination codon of ORF1 results in the 182 kDa large replicase subunit (ORF2). The ORF3 and ORF4 translate from 3' subgenomic RNAs and result in a 30 kDa movement protein, and a 17 kDa coat protein, respectively [45]. Several reports demonstrated that besides its role in replication, the p125 protein acts as a viral silencing suppressor through the binding of vsiRNAs [48–50]. We also observed this ecotype-specific leaf deformation in the case of turnip mosaic virus (TuMV) infection. TuMV is a member of the *Potyviridae* family and it is the most important virus infecting cruciferous crops [51]. The TuMV genome is around 9.8 kb and the genomic RNA is translated into a large polyprotein and a frame-shift protein. The large polyprotein is subsequently processed by the action of viral-encoded HCPro protease into ten mature functional products [52, 53]. A frame-shift protein, P3N-PIPO, was reported to be involved in the pathogenesis and movement of TuMV [54, 55]. The TuMV HCPro protein has RNA silencing suppressor (RSS) activity [56] through the binding of vsiRNAs [57, 58]. The ecotype-specific leaf deformation phenotype observations with two different RNA virus infections suggested that there are genetic differences in the hosts that determine this response. We also expected that this differential regulation will manifest in the transcriptome or sRNAome profiles and by comparing these profiles we will be able to find the host factors responsible for the ecotype-specific symptom development. Indeed, the comparative transcriptome analysis of the TVCV-, TuMV-, and mock-infected Col and Bur plants resulted in a few candidate genes that could be responsible for the leaf deformation. Some of these genes are known regulators of leaf polarity while others may represent novel, ecotype-specific functions in leaf development or other biological processes that are commonly disturbed by TVCV and TuMV infection.

## Materials and methods

### Plant growth conditions and viral infection

Seeds of *Arabidopsis thaliana* Col and Bur ecotypes were sown on soil in pots, watered, wrapped with cling film, and kept in the dark for two days. After that, pots were moved into a growth chamber (SANYO/Panasonic MLR-352-PE) where plants grew under long-day conditions (16 h light, 8 h dark, constant 21˚C) for two weeks. Then, individual seedlings were

transferred into Jiffies ([www.jiffypot.com](www.jiffypot.com)) and grown for further two weeks in the same chamber. The four-week-old seedlings were transferred into pots filled with soil and moved into a lightroom where they grew for one day under long-day conditions. The next day, in the case of TVCV-ApH, plants were inoculated as described before [45]. Briefly, 2 μL (1 μg) of either total RNA purified from systemic leaves of *Nicotiana benthamiana* infected with an infectious *in vitro* transcript of TVCV-ApH or water (mock) was combined with 5.5 μL water and 7.5 μL 2× inoculation buffer and was rub-inoculated into the true leaves of four-week-old *Arabidopsis* plants. In the case of TuMV, *Arabidopsis* plants were infected with extracts of *N. benthamiana* leaves infected with an infectious *in vitro* transcript of TuMV. For both viruses, three leaves were inoculated per plant and 60 plants were inoculated per sample (Col mock, Col virus, Bur mock, and Bur virus). After inoculation, the plants were sprayed with water and grown under long-day conditions in the same lightroom.

## Purification of RNA samples for high-throughput sequencing

For total RNA purification, two symptomatic systemic leaves were collected from every infected plant 14 day-post-inoculation (dpi) between ZT4 and ZT6. For the mock samples, leaves of similar age as the infected ones were collected. Total RNA was purified from individual leaves (480 samples altogether) using a phenol-chloroform extraction method [59]. The purified RNA samples were quantified by Nanodrop photometer and 1 μg of total RNA per sample was run on a denaturing agarose gel. The gel was stained with ethidium bromide and photographed with ChemiDoc™ (Bio-Rad, Hercules, CA, USA). The relative amount of the viral RNAs that were visible over the ribosomal RNA bands was quantified with Image Lab v5.1 software (Bio-Rad, Hercules, CA, USA) using the 18S rRNA bands as the reference. Seven samples with similar levels of viral genomic RNAs were pooled together to form a biological replicate. In this way, four replicates were created.

## High-throughput sequencing of mRNA and sRNA populations

The libraries for the RNA-seq and sRNA-seq from the TVCV-infected samples were prepared and sequencing reactions were carried out by LC Sciences LLC (Houston, TX, USA). The polyA-selected RNA-seq libraries were prepared using the TruSeq Stranded mRNA Library Prep Kit (Illumina, San Diego, USA) and sequenced with a paired-end 2×150 bp chemistry on Illumina HiSeq X platform. The sRNA-seq libraries were prepared using the TruSeq Small RNA Sample Prep Kit (Illumina, San Diego, USA) and sequenced on an Illumina HiSeq X platform with a single-end 50 bp chemistry.

PolyA-selected, stranded RNA-seq libraries for the TuMV samples were prepared and sequenced by Novogene Co., Ltd. (Cambridge, UK) on a NovaSeq 6000 sequencing platform with a paired-end 2×150 bp chemistry. The sRNA libraries (including a size-separation by PAGE) were prepared in our laboratory using the TruSeq Small RNA Sample Prep Kit (Illumina, San Diego, USA) and were sequenced by Novogene Co., Ltd (Cambridge, UK) on a NovaSeq 6000 platform with a single-end 50 bp chemistry.

The raw sequencing data were deposited in the NCBI SRA database (BioProject: PRJNA788379) under the identifiers SRR17227515–SRR17227578.

## Bioinformatic analysis of high-throughput sequencing data

**RNA-seq analysis.** The quality of the raw reads was checked with FastQC v0.11.8 [60]. Trimming of adapter sequences and filtering of low-quality reads were carried out with cutadapt v2.8 [61]. The clean reads were aligned to the AtRTD2 reference transcriptome [62] and the normalized expression values (transcript per million, TPM) were calculated with kallisto

v0.44.0 [63] with the following parameters: -b 10—bias—rf-stranded. The TPM values calculated for the alternative transcript isoforms were aggregated to represent gene-level expression and were normalized between samples with sleuth v0.29.0 [63]. To test whether there is a significant difference between the means of the Bur (or Col) virus-infected and all the other samples, a Wald test was applied. The calculated *P*-values were corrected for multiple testing using the Benjamini-Hochberg method [64]. Genes with *qval* (corrected *P*-value, at 1% false discovery rate) < 0.01, *b* (estimated fold-change) > log(2), and *mean_obs* (mean raw expression level) > 1 were considered as differentially expressed genes (DEGs).

The DEG expressions were visualized in a heatmap, for which *Z*-scores were calculated in the following way: for every gene, the mean and standard deviation of the expression across samples were calculated, then from every individual value the mean was subtracted and the resulting value was divided by the standard deviation. The *Z*-score tells how many standard deviations an expression value is from the mean expression of an individual gene. In this way, genes with different mean expression levels can be compared. The heatmaps were generated with the geom_tile function of the ggplot2 R package [65].

Principal component analysis of the RNA-seq data was performed using the prcomp R package. Normalized gene expression values (transcript per million, TPM) were scaled (log2-transformed) and centered before the analysis. The plots were generated with the autoplot function of the ggfortify R package [66].

Comparison of the DEG sets from different conditions was analyzed with the UpSet R package [67].

The Gene Ontology (GO) term analysis of the DEGs was performed using the PlantRegMap server [68] with the default settings. The enrichment factor was calculated by dividing the observed number of genes in a particular category by the expected number of genes in that category. The plot was generated with the geom_point function of the ggplot2 R package [65].

## Small RNA-seq analysis

The raw sequences were processed and filtered using cutadapt v2.8 [61] with the following parameters: -a TGGAATTCTCGGGTGCCAAGG -m 20 -M 25 -q 20—max-n = 0—discard-untrimmed. The trimmed sequences were further filtered to remove tRNA- and rRNA-derived sequences using the short read aligner bowtie2 v2.3.5.1 with the following parameters: -k 1 -D 20 -R 3 -N 1 -L 10 -i S,1,0.50 and a reference sequence set containing tRNA, rRNA, snRNA, and snoRNA sequences from *Viridiplantae*. The unaligned sequences were collected and aligned to the *Arabidopsis thaliana* TAIR10 reference genome with ShortStack v3.8.5 [69] with the following parameters:—align_only—bowtie_m all—ranmax none—keep_quals. The raw count table of the genome-mapped sRNA sequences was produced using the fasta and view functions of the samtools suite v1.9 [70] and custom shell scripts. The sRNS-producing loci were predicted using ShortStack v3.8.5 with the following parameters:—dicermin 20—dicermax 25. The normalization and differential expression analysis of the sequences or the sRNA loci were performed with the DESeq2 R package [71]. Only those sequences/loci were accepted that passed through the following filter: *padj* (corrected *P*-value, at 1% false discovery rate) < 0.05, *log2FoldChange* > log(1.5), and *baseMean* (mean raw expression level) > 1. The DESeq2 normalization factors were used for the calculation of the sequence length distribution plot.

The sRNA target analysis was performed using the psRNATarget server [72] with the default settings except that the target accessibility calculation was allowed. We provided the 114 Bur virus upregulated sRNAs and the 88 Bur virus downregulated transcripts that were extracted from the AtRTD2 transcripts [62] using the notseq function of the EMBOSS package v6.6 [73]. Only the predicted cleavage hits were considered for further analysis.

## RT-qPCR analysis

The primers to measure *ARF4* (AT5G60450) expression (ARF4_qPCR_F: 5'- GCT CCT CTT GAC TAC AAA CAA CAG -3', ARF4_qPCR_R: 5'- GGC GAA ACT TCC ACT CTA CTC C -3') were designed with PerlPrimer v1.2.4 [74] in a way that at least one primer must span an intron to prevent amplification from genomic DNA. The *FIL* (AT2G45190) was amplified using the primers described before (FIL_qPCR_F: 5'- TGG TAC AGC AAC CAC ATC GGA CAG -3', FIL_qPCR_R: 5'- GCC AAA CCA TCC TTG CGG TTA ATG -3') [75]. The *PDF2* gene (AT1G13320) was used as an internal reference, primers to measure *PDF2* were published earlier (PDF2_qPCR_F: 5'- TCA TTC CGA TAG TCG ACC AAG -3', PDF2_qPCR_R: 5'- TTG ATT TGC GAA ATA CCG AAC -3') [76].

For the first-strand cDNA synthesis, 1 μg of total RNA was treated with DNase I, then reverse transcribed with RevertAid H Minus Reverse Transcriptase (Thermo Fisher Scientific, Waltham, MA, USA) and a random primer according to the manufacturer's instructions. The cDNAs were diluted ten times, and 1 μL was used in a 10 μL qPCR. The qPCRs were performed using SYBR™ Select Master Mix (Thermo Fisher Scientific, Waltham, MA, USA), following the manufacturer's instructions. The reactions were run in a LightCycler® 96 Real-Time PCR machine (Roche, Basel, Switzerland). The reaction profile was the following: preincubation at 95°C for 1 min; amplification at 95°C for 15 s and 60°C for 30 s, repeated 45 times; melting at 95°C for 10 s, then 65°C for 1 min and continuous heating to 97°C; cooling at 37°C for 30 s. Four biological and two technical replicates were measured per sample. The values for every biological replicate were calculated as the mean of the two technical replicates. The relative expression values were calculated by the LightCycler96® v1.1 software (Roche, Basel, Switzerland). Unpaired, two-tailed Student's *t*-tests (at 0.95 confidence level) were performed using the t.test R package to assess if there is a significant difference between the means of the mock- and virus-infected samples in the Col and Bur plants either in the TVCV or TuMV series.

## QTL mapping

To obtain a coarse genetic map, a publicly available Minimal Recombinant Inbred Line set (INRA, Versailles, France) derived from a cross of *Arabidopsis thaliana* Bur and Col ecotypes was amended with 23 more lines selected randomly from the remaining lines [77]. Altogether, 43 lines and the two parental lines were used in the experiment. Five plants per line were infected either with TVCV-ApH or mock as described above and the leaf deformation (upward curling) was scored from 0 to 5 (0 = infected Col, 5 = infected Bur). The average score of the five plants was recorded and used for assessing correlation with the genotypes of the RILs that were obtained from INRA (http://publiclines.versailles.inrae.fr/page/20). The analysis was carried out with Windows QTL Cartographer v2.5_011 [78] using the single marker association test.

## Results

### TVCV and TuMV infection causes leaf deformation only on Bur ecotype

We observed earlier that upon infection with turnip vein-clearing virus (TVCV, a tobamovirus), the Bur ecotype of *Arabidopsis thaliana* displayed much more severe symptoms than Col [45]. The most obvious symptom was the leaf deformation which included polarity defect (upward curling leaves) and wavy leaf margin that could not be observed on Col plants under the conditions tested (Fig 1). We were curious whether this difference in symptom development is specific to this virus-host interaction or can be observed with other viruses as well.

Therefore, we infected the above-mentioned two *Arabidopsis* ecotypes with turnip mosaic virus (TuMV, a potyvirus). We observed the same difference in symptom development between the two ecotypes, namely, the Bur plants showed upward curling leaves while the Col plants did not have leaf deformity (Fig 1).

Both viruses replicated and reached similar viral RNA levels in each ecotype, suggesting that viral replication, short-distance, and long-distance viral movement are not involved in the observed symptom differences. The differential leaf development defect in the two ecotypes upon both virus infections suggests that there are host factors whose ecotype-specific misregulation leads to an altered leaf developmental program only in the Bur plants.

## TVCV and TuMV infection elicit different transcriptome responses in Col and Bur ecotypes

Upon viral infection, the transcriptome of the infected host changes significantly [79–83]. The virus modulates the host environment in favor of its replication and spread, while the host tries to restrict the intruder and initiates an immune response. We expected that the two ecotypes would respond to the virus similarly in many ways. However, since there were significant differences in the observable symptoms, we anticipated that there would be gene expression changes that reflect these differences. In order to reveal these differential gene expression profiles, we collected leaves from TVCV- and TuMV-infected *Arabidopsis* Col and Bur plants along with mock-infected ones in four biological replicates for RNA-seq experiments and purified total RNA from them. Next, the polyA-tail mRNA fraction was sequenced either on an Illumina HiSeq X (TVCV) or a NovaSeq 6000 (TuMV) platform. The clean reads were mapped to the AtRTD2 *Arabidopsis thaliana* reference transcriptome [62] with kallisto [63] and the gene expression values were normalized between samples with sleuth [63] (S1 Table). We performed a principal component analysis to reveal which experimental variables are responsible for the most variances between samples (Fig 2).

According to the analysis, the most important factor that separates samples and is responsible for 38.23% of the total variances (PC1) is the ecotype, while the second most important factor (PC2) separates the TVCV and TuMV series (Fig 2A). This latter is either because of the different technology that was used for sequencing the two series in two different laboratories or could reflect a true biological difference (or both). The third most important factor (PC3) that separates the mock- and virus-infected samples is responsible only for 8.06% of the total variances observed between samples (Fig 2B). This figure also shows that the mock- and virus-infected samples in the Bur ecotype are more separated than in the Col ecotype (Fig 2B). We interpret these results that there is a substantial difference between the two ecotypes in their responses to viral infection.

## Finding the genes responsible for the common leaf polarity defect in the TVCV- and TuMV-infected Bur plants

To find out which genes are responsible for the upward curling leaves in the TVCV- or TuMV-infected Bur ecotype, we performed a differential gene expression analysis using sleuth [63], separately for the two virus series. We expected that genes responsible for the leaf deformation that was observed only in the virus-infected Bur ecotype will either be induced or reduced only in the Bur virus samples both in the TVCV and TuMV series compared to all the other samples in the series, namely, Bur mock, Col mock and Col virus samples. Alternatively, it is possible that genes are preferentially induced or reduced in the Col virus samples that prevent symptom development. We considered genes differentially expressed only if the *Q*-value from the statistical test was lower than 0.01 (1% false discovery rate), the absolute value of the effect (*b* parameter, or estimated

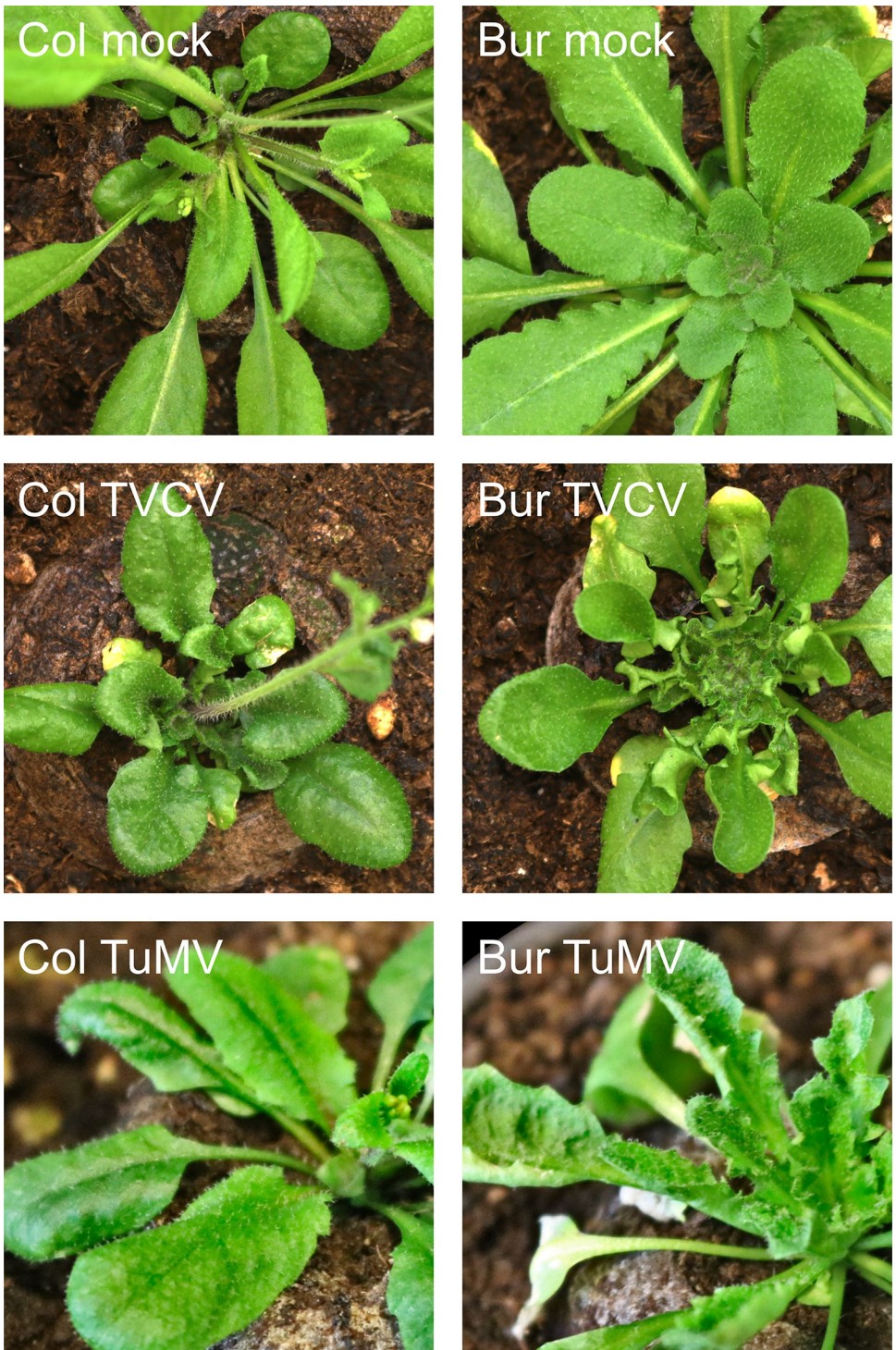

**Fig 1. Viral symptoms of TVCV- and TuMV-infected *Arabidopsis thaliana* Col and Bur ecotypes.** Four-week-old plants grown under long-day conditions at 21˚C were infected either with virus or mock solutions. After two weeks, plants were photographed and samples for RNA purifications were collected from mock- and virus-infected plants. For the RNA purification, samples were taken from symptomatic leaves of Bur or leaves of similar developmental stages from the Col and mock samples. To obtain one replicate, seven leaves from different plants were pooled (one leaf per plant). Four replicates were prepared in such a way for every mock and virus-infected sample.

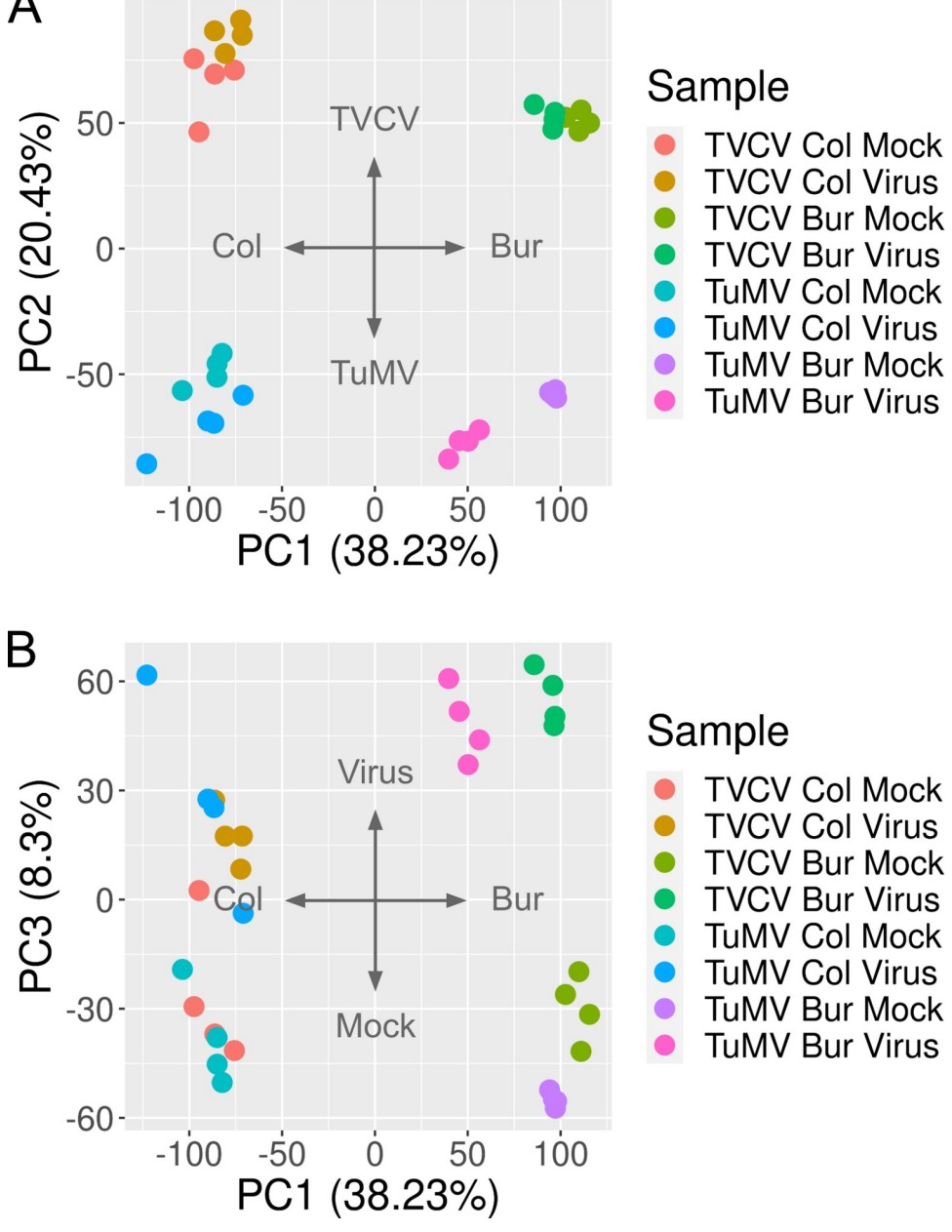

**Fig 2. Principal component analysis of the RNA-seq data.** PCA was performed using the prcomp R package. Normalized gene expressions (Transcript Per Million, TPM) were scaled and centered before the analysis. (A) The first component (PC1) separates the samples by ecotype and explains 36.85% of the total variances between the samples while the second component (PC2) separates the samples by virus series and explains further 23.35% of the variances. (B) The third component (PC3) separates samples that are either mock or virus-infected, which is responsible for a further 8.06% of the variances. The four biological replicates are marked with the same color for easier identification.

fold-change, log-transformed) was higher than log(2), and the mean raw expression (*mean_obs*) was higher than 1. In this way, we got a list of the differentially expressed genes for the Bur virus or Col virus samples for the TVCV and TuMV series (Fig 3).

There were 278 genes that were up- and 548 that were downregulated in the Bur virus samples, while 386 and 162 genes were up- and downregulated in the Col virus samples in the

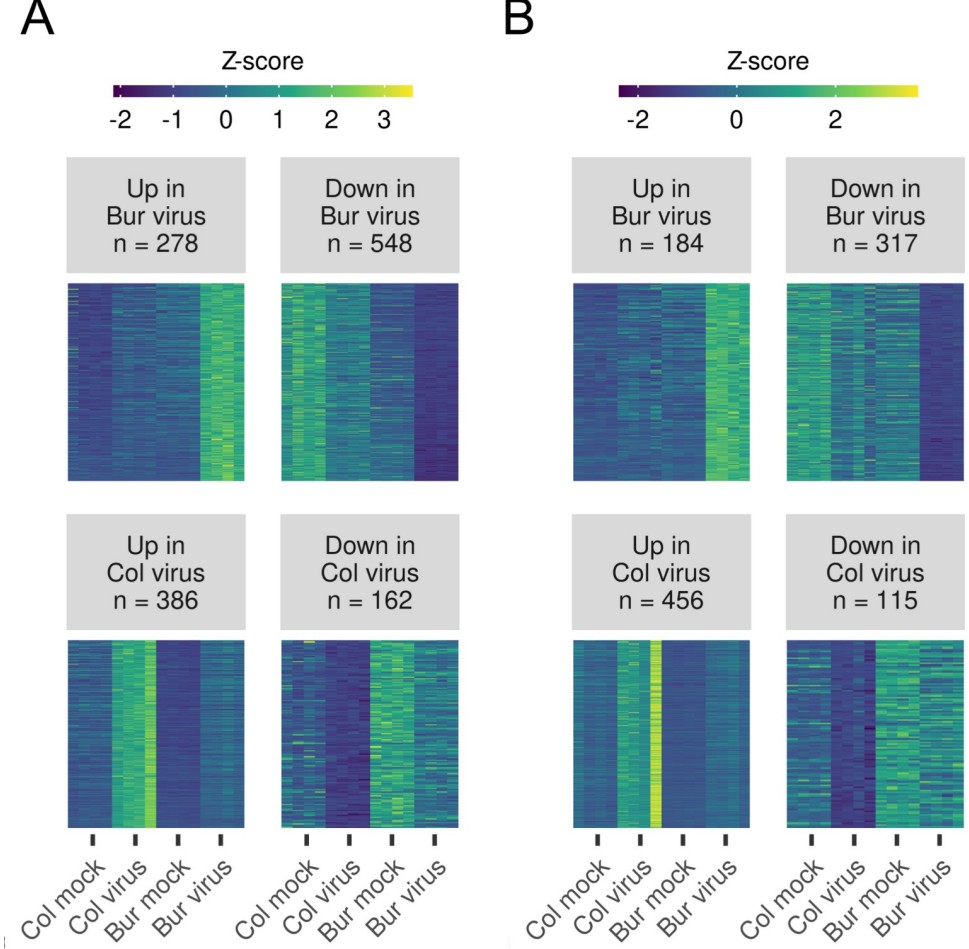

**Fig 3. Expression pattern of the differentially expressed genes between Bur and Col plants infected either with TVCV or TuMV.** To get a list of differentially expressed genes (DEGs) that significantly change only in the Bur (or Col) virus-infected samples, a Wald-test was applied (i.e., Bur virus samples against all the others). Only genes with $qval < 0.01$, $b$ (estimated fold-change) $> \log(2)$ and $mean\_obs$ (mean raw expression level) $> 1$ were accepted as DEGs. The number of DEGs is noted in the title. Normalized expression values (transcript per million, TPM) of individual, differentially expressed genes in (A) TVCV and (B) TuMV-infected samples were centered by calculating Z-scores. Z-score tells how many Standard Deviations an expression value is from the mean expression of an individual gene. In this way, genes with different mean expression levels can be compared. The expressions of DEGs in samples are represented in a heatmap. The expressions of the four biological replicates are shown for every sample.

TVCV series (Fig 3A, S2 Table). These numbers were 184, 317, 456, and 115 in the TuMV series, respectively (Fig 3B, S2 Table). To filter for the genes that were regulated similarly in the two virus series, we performed a set analysis with the UpSet R package (Fig 4).

According to this analysis, the majority of the differentially expressed genes were unique to a certain set and probably not responsible for the commonly observed leaf polarity misregulation. The most abundant intersection with 88 genes was the one that contained downregulated genes in Bur virus samples both in the TVCV and TuMV series (S3B Table), while the second most abundant with 36 genes was the one containing the upregulated genes in Bur virus samples in both virus series (S3A Table). All the other intersections were much smaller and not relevant for our investigation. We performed a GO term enrichment analysis with the above-mentioned two gene sets using the PlantRegMap server [68]. Among the upregulated genes,

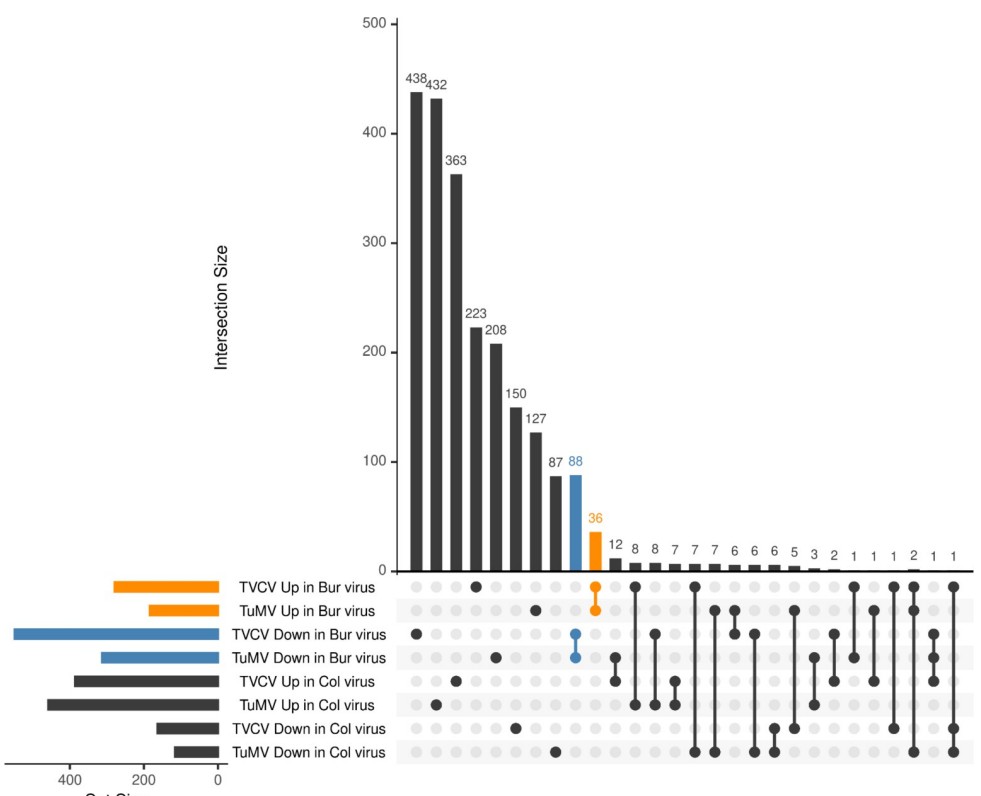

**Fig 4. Commonly regulated genes in TVCV and TuMV-infected plants.** Sets of DEGs in eight different conditions were compared and visualized using the UpSet R package. The vast majority of the DEGs were specific to one condition (represented by single dots), while the number of common elements between the different conditions (connected dots) was much smaller. The most abundant common elements were observed between the sets of downregulated genes in Bur virus samples both in TVCV and TuMV-infected plants (blue) and the upregulated genes in Bur virus samples both in TVCV and TuMV-infected plants (orange). All the other common elements were much smaller in size, sets with zero elements are not shown.

categories were enriched with moderate or low significance, related to plant cell wall functions (GO:0071554, GO:0071555, GO:0071669, etc), cellular developmental process (GO:0048869), response to hormone (GO:0009725), cell differentiation (GO:0030154) while among the downregulated genes, several terms related to photosynthesis and light regulation (GO:0009768, GO:0009765, GO:0019684, etc), hormone metabolic process (GO:0042445), regulation of hormone level (GO:0010817), regionalization (GO:0003002), pattern specification process (GO:0007389), etc., were significantly enriched (Fig 5, S3C and S3D Table).

Interestingly, the latter two categories (GO:0003002: regionalization and GO:0007389: pattern specification process) in the downregulated genes contained *FIL* (*FILAMENTOUS FLOWER*, AT2G45190) that encodes a member of the YABBY family of transcriptional regulators involved in abaxial cell type specification in leaves [84]. Actually, *FIL* is the most significantly differentially down-regulated (in the Bur virus samples) gene in the common set of the TVCV and TuMV series (S3B Table). *FIL* directly regulates *ARF4 (AUXIN RESPONSE FACTOR 4*, AT5G60450)* [85] that was found in the response to hormone and the cellular developmental process GO categories in the enriched sets of upregulated genes. Furthermore, the important leaf abaxial identity determinant *ARF4* is known to be post-transcriptionally regulated by tasiARF-guided AGO1, as described in the Introduction part.

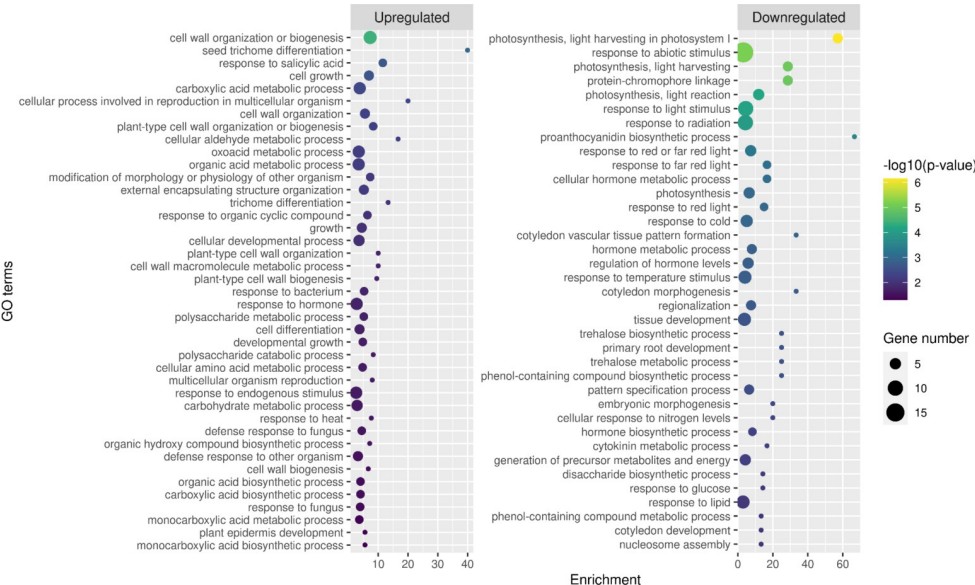

**Fig 5. GO term enrichment analysis of common DEGs of TVCV and TuMV-infected Bur plants.** The common DEGs of the TVCV and TuMV-infected Bur plants (orange and blue colored sets in Fig 4) were subjected to GO-term enrichment analysis using the PlantRegMap server with the default settings. Only terms in the Biological Process category are shown, the results of the full analysis can be found in S3C and S3D Table. Enrichment was calculated by dividing the observed number of genes in a particular category by the expected number of genes in that category.

## Virus-responsive sRNA profiles of the ecotypes are different

To profile the small RNA landscape upon virus infection and match it with the transcriptome profiles, we sequenced the sRNAome in the same samples that were used for the RNA-seq experiments. We performed a Principal Component Analysis of the normalized sRNA expression values to check the quality of our dataset (Fig 6).

According to the PCA, the first principal component separates samples by the virus series which could either mean that the two viruses elicit quite different responses or could reflect the different sample preparation and sequencing platforms that were used for the generation of data (or both). The second principal component separates the samples by ecotype, while the third one by treatment. The substantial differences in the responses of the two ecotypes to the two viruses suggested that there would be a limited number of commonly regulated sRNAs that could be responsible for the observed common symptoms.

Next, we analyzed the length distribution of the sRNAs in the samples. Normally, the 24-nt sRNAs dominate the sRNA landscape, while the second more abundant size class is the 21-nt sRNAs. Upon virus infection, this size distribution switches: the 21–22-nt size classes become overwhelmingly dominant [13, 44, 86, 87]. This is mainly due to the 21- and 22-nt viral siR-NAs that are produced by DCL4 and DCL2 enzyme activities, respectively, and amplified by RDR1 or RDR6 in *Arabidopsis* [5, 88–90]. Analysis of our sRNA libraries showed the same shift in size classes. However, when we filtered out the virus-derived vsiRNAs by mapping the sRNA sequences to the *Arabidopsis* genome, the remaining host-derived sRNAs also showed a tremendous 21-nt size class enrichment in the virus-infected samples (Fig 7).

This can either be attributed to the mostly 21-nt-long miRNAs or host-derived, also vastly 21-nt-long virus-activated siRNAs (vasiRNAs), which are produced in an RDR1 and DCL4-dependent manner from plant precursor transcripts [42]. The vasiRNAs regulate host genes that can modulate the antiviral response of the plant. We hypothesized that some

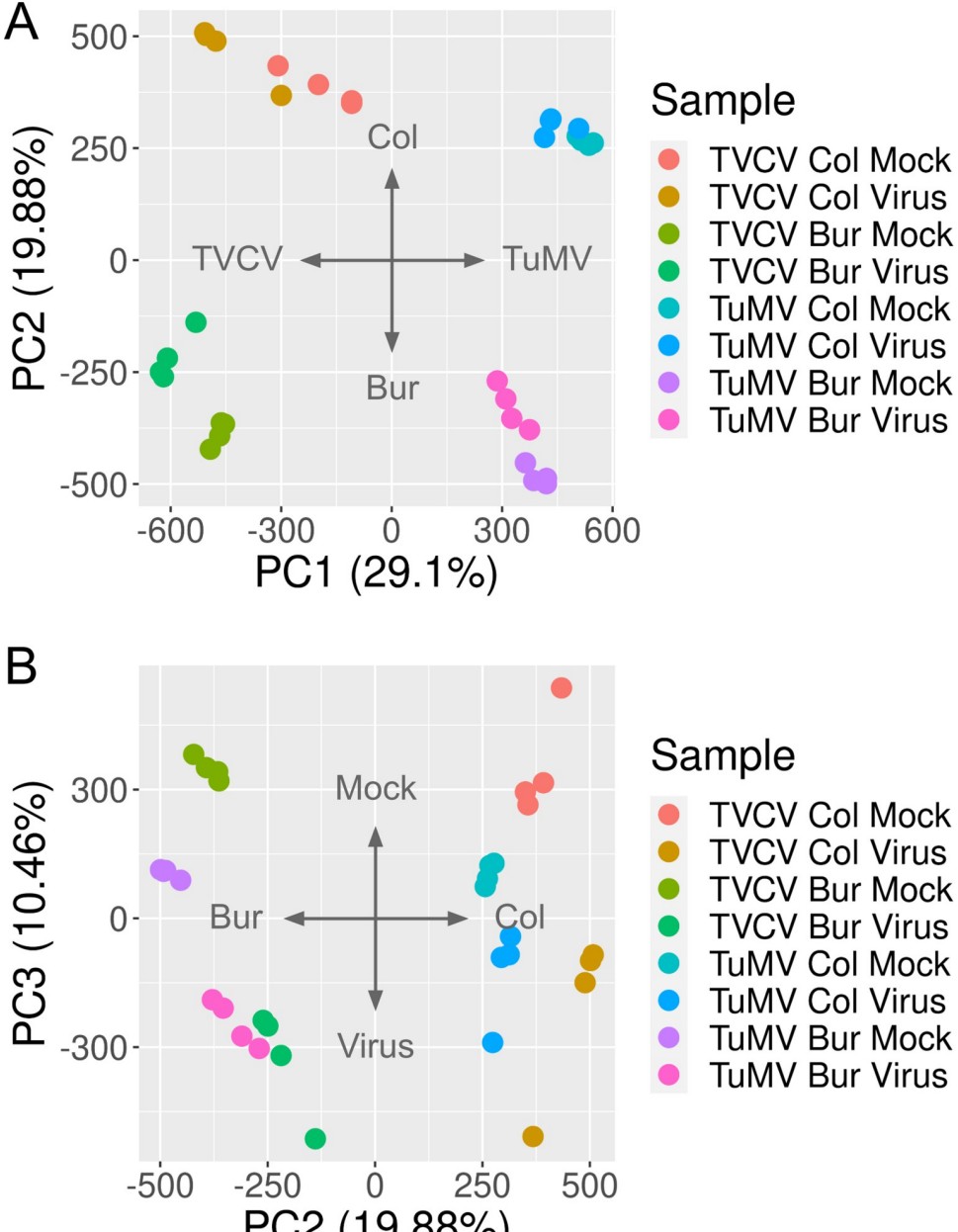

**Fig 6. Principal component analysis of the sRNA-seq data.** PCA was performed using the prcomp R package. DESeq2-normalized sRNA expressions were scaled and centered before the analysis. (A) The first component (PC1) separates the samples by virus series and explains 29.1% of the total variances between the samples while the second component (PC2) separates the samples by ecotype and explains further 19.88% of the variances. (B) The third component (PC3) separates samples that are either mock or virus-infected, which is responsible for a further 10.46% of the variances. The four biological replicates are marked with the same color for easier identification.

vasiRNAs may be produced in an ecotype-specific manner and can regulate the genes that we have found in the comparative transcriptome analysis. To find sRNAs whose expression either increased or decreased only in the Bur virus samples, we performed a differential expression analysis of the genome-mapped, clean sRNAs which had a mean read count > 1 (102195 unique, 20–25-nt-long sequences) using DESeq2. We applied a filtering rule that was less strict

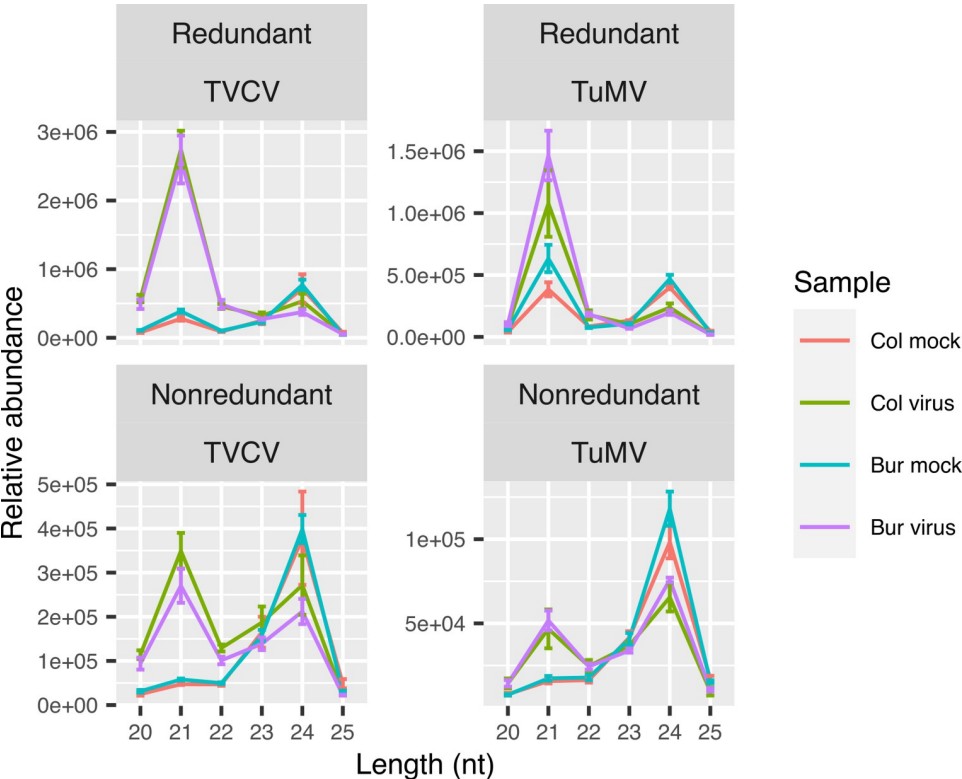

**Fig 7. Sequence-length distribution of the host-derived sRNAs in Bur and Col plants infected either with TVCV or TuMV.** The raw sRNA sequences were first adapter-trimmed, quality- and length-filtered, then rRNA and tRNA sequences were removed. The remaining sRNA sequences were mapped to the *Arabidopsis thaliana* TAIR10 reference genome with ShortStack. The raw abundances of the genome-mapped sequences were normalized with DESeq2. The normalization factors were used to scale the raw abundances of the size classes measured by FastQC. Error bars represent the standard error of the mean of four biological replicates.

than in the case of RNA-seq analysis (log2FoldChange > log2(1.5), padj < 0.05, baseMean > 0) because according to our experience, sRNA expressional changes are usually smaller and the data are noisier. In this way, we got sets of differentially expressed sRNAs in the Bur (or Col) virus samples in the TVCV and TuMV series (Fig 8).

The analysis revealed that sRNAs tended to be upregulated upon virus infection rather than downregulated: there were 2266 sequences that were selectively upregulated and 38 downregulated in the Bur virus samples in the TVCV series, and 521 and 48 in the TuMV series. These numbers were similar regarding the selectively up- and downregulated sRNAs in the Col virus samples in both virus series (Fig 8). The majority of upregulated sRNAs in both virus series were 21–22-nt-long while the downregulated sRNAs were mainly 23–24-nt-long (S4 Table).

To find commonly regulated sRNAs between the TVCV and TuMV series, we analyzed the=differentially expressed sRNAs with the UpSet R package like in the case of RNA-seq data (Fig 9).

We focused on the sets that have common sequences between the TVCV and TuMV Bur virus samples, either up- or downregulated to find potential candidates that can regulate the DEGs identified in the RNA-seq analysis. We found 133 upregulated and one downregulated sRNA sequence that was common between the TVCV and TuMV Bur virus samples. Annotation of these sequences revealed that most of the 133 upregulated sequences are 21-nt long and associated with intergenic regions, transposable elements, *TAS1*, and *TAS2* genes, some

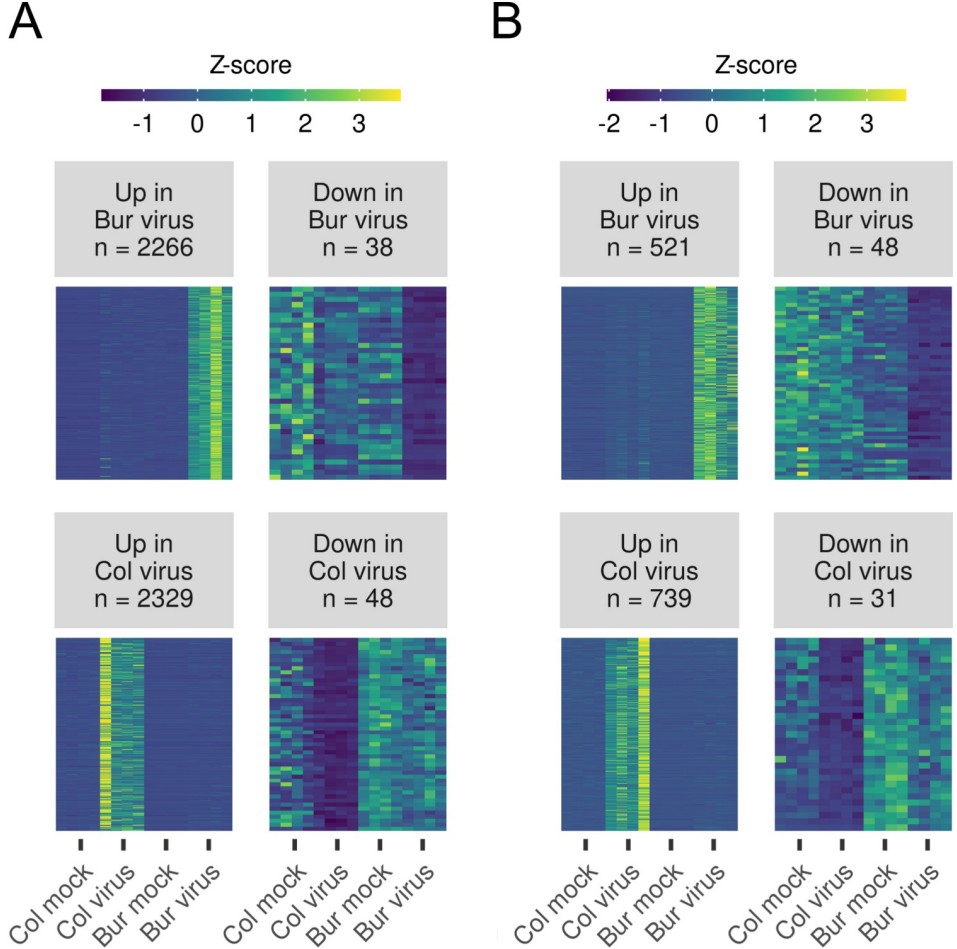

**Fig 8. Expression pattern of the differentially expressed sRNA sequences between Bur and Col plants infected either with TVCV or TuMV.** To get a list of differentially expressed sRNAs (DESs) that significantly change only in the Bur (or Col) virus-infected samples, a Wald-test was applied (i.e., Bur virus samples against all the others). Only sequences with *padj* < 0.05, *Log2FoldChange* > log(1.5), and *baseMean* (mean raw expression level) > 0 were accepted as DESs. The number of DESs is noted in the title. Normalized expression values of the individual, differentially expressed sequences in (A) TVCV and (B) TuMV-infected samples were centered by calculating *Z*-scores. *Z*-score tells how many Standard Deviations an expression value is from the mean expression of an individual sequence. In this way, sequences with different mean expression levels can be compared. The expressions of DESs in samples are represented in a heatmap. The expressions of the four biological replicates are shown for every sample.

protein-coding genes and pseudogenes, and some miRNA genes (*MIR158B*, *MIR162B*, *MIR390A*, *MIR396A*, *MIR835A*, *MIR841A*, *MIR846A*), although only one mature miRNA sequence was found (ath-miR846-3p), all the other MIR-associated sequences are iso-miRs or siRNAs. It is worth mentioning that 41% of the sequences contain one mismatch (including the *TAS1C*- and *TAS2*-derived, and miR390, miR396, miR835a, miR841a sequences) which suggests that they are Bur-specific sequences. We checked if the 133 Bur virus upregulated sequences can be found in the Col virus upregulated sequences allowing 2 mismatches and we found that 19 sequences were also present in the Col set, so we removed them. We were curious if the remaining 114 Bur virus upregulated sRNAs can target the 88 Bur virus downregulated genes identified in the RNA-seq analysis, therefore, we performed a target prediction using the psRNATarget server [72]. According to the prediction, 86 out of the 114 sRNAs can potentially regulate 69 out of the 88 genes (S5 Table). When we performed a GO term

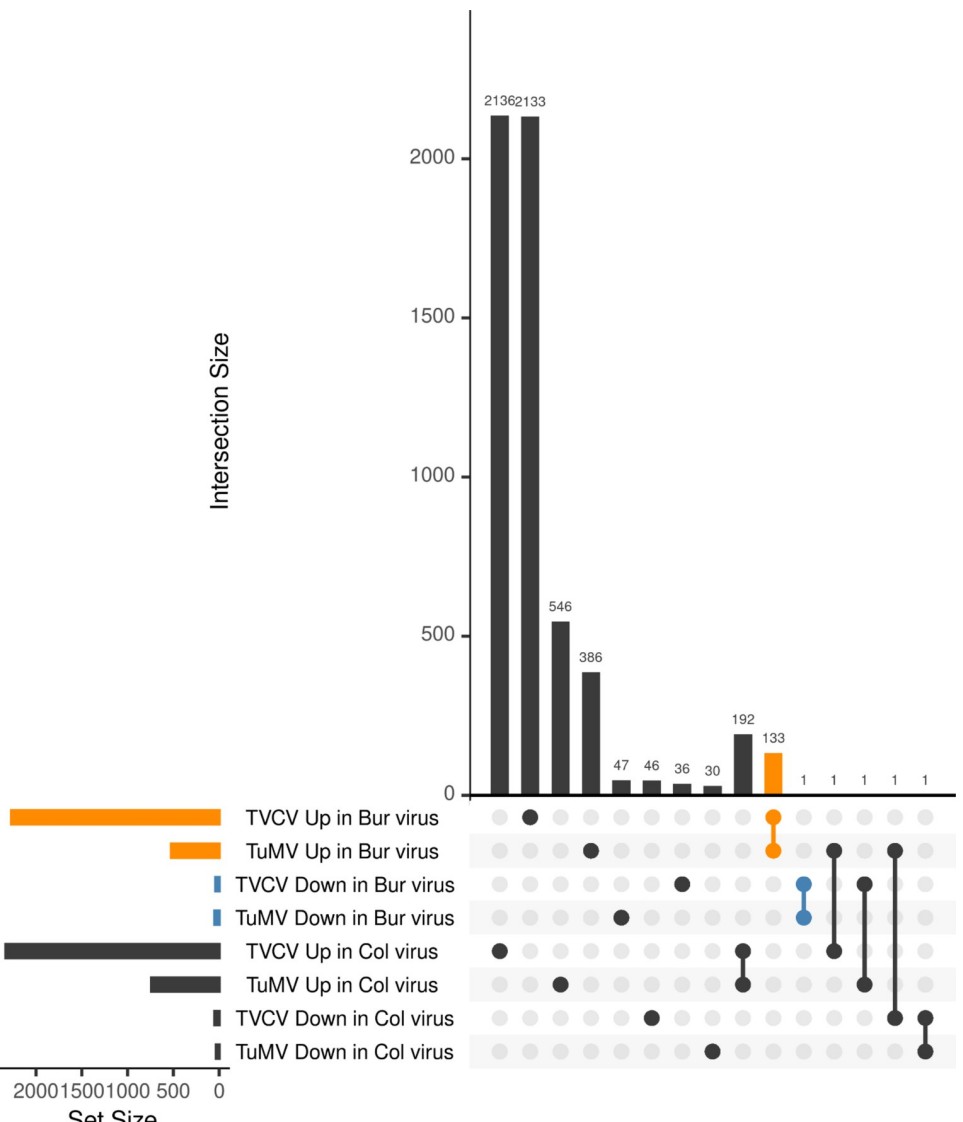

**Fig 9. Commonly regulated sRNA sequences in TVCV and TuMV-infected plants.** Sets of differentially expressed sRNA sequences (DES) in eight different conditions were compared and visualized using the UpSet R package. The vast majority of the DESs were specific to one condition (represented by single dots), while the number of common elements between the different conditions (connected dots) was much smaller. Blue color marks the sRNA sequences that are downregulated in Bur virus samples both in TVCV and TuMV-infected plants while orange color marks the sRNA sequences that are upregulated in Bur virus samples both in TVCV and TuMV-infected plants. Sets with zero elements are not shown.

enrichment analysis of the potentially vasiRNA-regulated genes, the regionalization and pattern specification categories remained significantly enriched (S1 Fig in S2 File). We found that a transposon-derived 21-nt-long siRNA (5'- AUU GCA GAG AUG GAU GUA CAA -3') could potentially target *FIL in trans*. The *FIL*-targeting sequence is identical in the two ecotypes but induced only in the Bur virus sample. One other gene in the GO categories regionalization (GO:0003002) and pattern specification process (GO:0007389) is also potentially regulated by a vasiRNA: *DOT3* (*DEFECTIVELY ORGANIZED TRIBUTARIES 3*, AT5G10250) is targeted by an intergenic siRNA that contains one mismatch compared to the Col reference sequence, but the Col version is not induced in the Col virus sample. The most significantly

enriched categories are related to photosynthesis and chloroplast functions. This is consistent with the previous observations that vasiRNAs target genes coding for different members of the photosynthetic apparatus to inhibit the replication of viruses by tuning down the energy production of the plant cell [43]. Apparently, there is an ecotype difference in this response that could contribute to the symptom severity.

The miR396 has been described to target the GRF family of transcription factors that are required for coordination of cell division and differentiation during leaf development in *Arabidopsis* [91]. The GRFs are not among the 88 downregulated genes we found. We checked their expression pattern in our data but they do not show any consistent pattern. There might be other genes related to leaf development directly or indirectly that could be regulated post-transcriptionally by vasiRNAs.

The one commonly downregulated sequence is the tasiARF that regulates *ARF4* which was found in the 36 Bur virus upregulated genes (S4 Table).

## tasiARF expression anti-correlates with the expression of its target *ARF4*

The finding that tasiARF is the only commonly downregulated sRNA between the TVCV and TuMV Bur virus samples prompted us to look up the expression profile of the tasiARF-related pathway components, namely, the phase-initiating miR390 that targets *TAS3* (the precursor of tasiARF), *AGO7*, the protein product of which binds miR390, *RDR6*, *SGS3*, *DRB4*, and *DCL4*, the protein products of which are involved in the synthesis and dicing of the dsRNA precursor, the tasiARF, and its target *ARF4*, the identified vasiRNA that potentially targets *FIL*, and *FIL* itself that together with *ARF4* are involved in the leaf polarity determination (Fig 10).

Analyzing Fig 10, we can observe the following: miR390 is virus-induced in both ecotypes, especially by TVCV infection. The *AGO7*, *RDR6*, and *DRB4* mRNA levels did not change significantly, while the *SGS3* was induced only in the Bur virus samples in both virus series. *DCL4* was induced in the Bur virus sample but the induced level was the same as the levels in the Col mock and virus samples. The tasiARF levels significantly drop in the Bur virus samples both in the TVCV and TuMV series. The expression profile of *ARF4* is anti-correlated with the

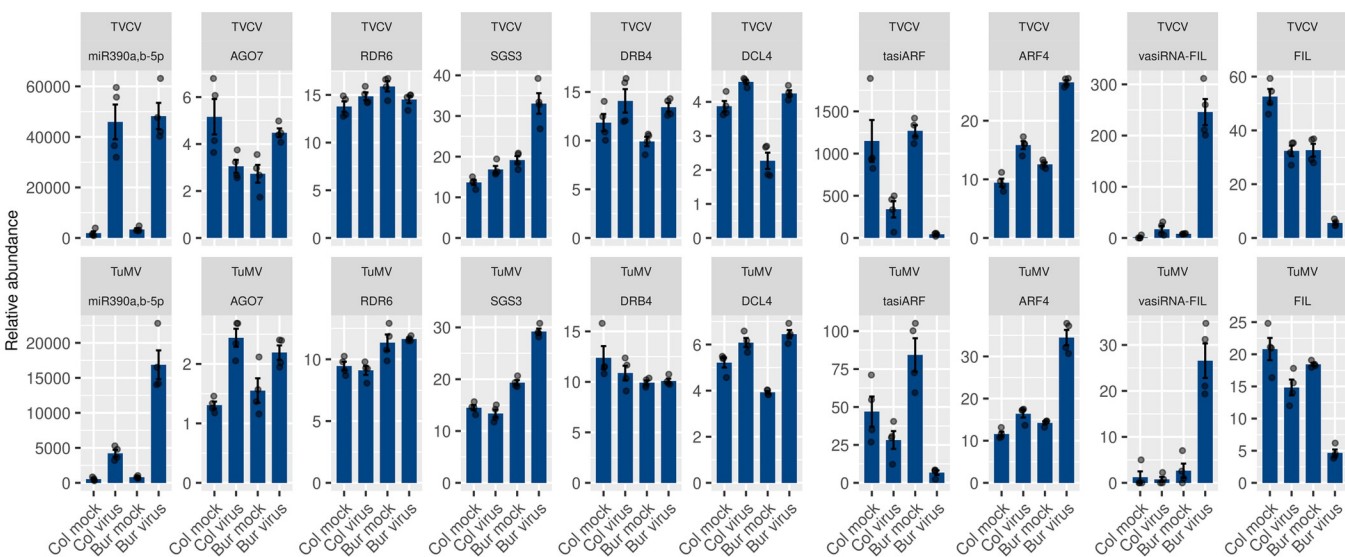

**Fig 10. Expression pattern of the *TAS3* regulatory module components.** The normalized expression values of the sRNAs and genes involved in the *TAS3* regulatory module are shown in the TVCV and TuMV series. Error bars represent the standard error of the mean of four biological replicates. The actual data points are also shown.

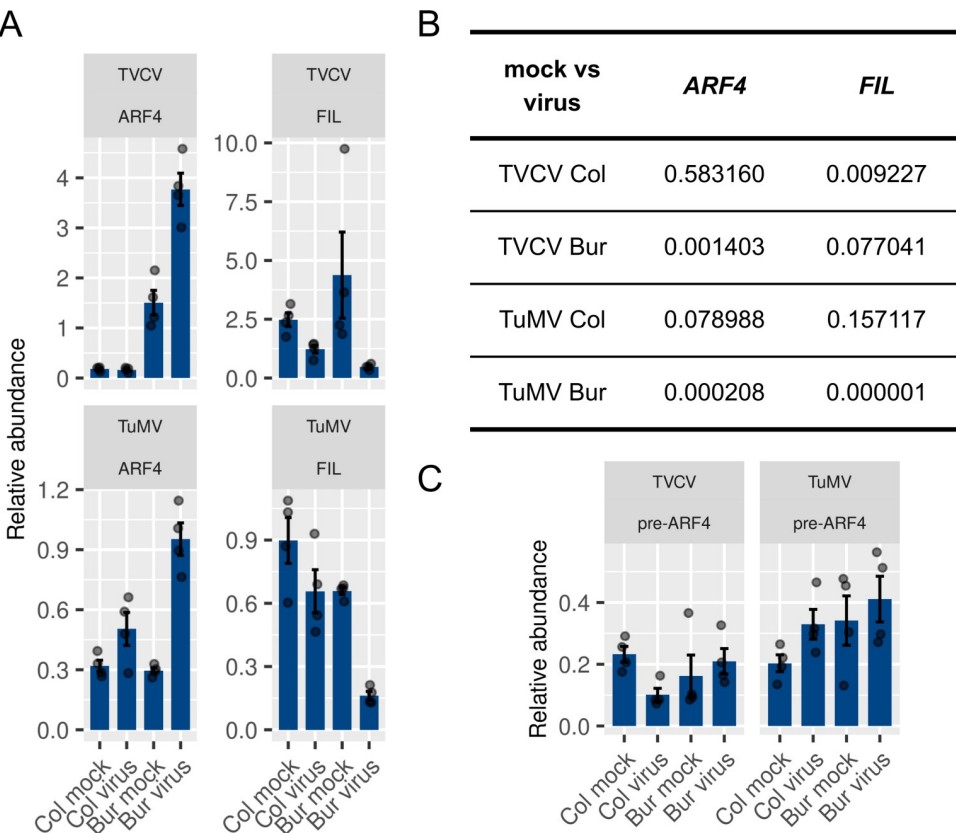

**Fig 11. Validation of the *ARF4* and *FIL* expression patterns with RT-qPCR.** (A) RT-qPCRs were performed on the samples used for the high-throughput sequencing. Four biological and two technical replicates were measured per sample. The values for every biological replicate were calculated as the mean of the two technical replicates and shown as dots in the figure. The error bars represent the standard error of the mean of four biological replicates. The *PDF2* gene (AT1G13320) was used as an internal reference [76]. (B) Unpaired, two-tailed Student's *t*-tests at 95% confidence intervals were performed to assess if there is a significant difference between the means of the mock and virus-infected samples in the Col and Bur plants either in the TVCV or TuMV series. The resulting *P*-values for every pair tested are indicated in the table. Values less than 0.05 indicate a significant difference. (C) Expression pattern of the unspliced pre-mRNA transcript of *ARF4* (genomic or *gARF4*) in the RNA-seq data. The error bars represent the standard error of the mean of four biological replicates. The individual data points are also shown.

tasiARF and the *FIL* profile in both virus series. The newly identified vasiRNA-FIL is dramatically induced in the Bur virus samples both the TVCV and TuMV series and anti-correlates with the *FIL* expression. The *ARF4* and *FIL* expression levels were validated by RT-qPCR (Fig 11).

To decide whether the observed elevated level of *ARF4* is due to a transcriptional (i.e., induction by a transcription factor like *FIL*) or post-transcriptional regulation (i.e., by tasiARF), we checked the expression level of the unspliced pre-mRNA transcript of *ARF4* by amending the reference transcriptome with the genomic sequence of the *ARF4* gene (*gARF4*) and performed an RNA-seq analysis as described before. According to this analysis, the *gARF4* transcript levels were very low in all samples and no significant change was observed that could explain the elevated level of *ARF4* in the Bur virus sample (Fig 11C). This suggests that the observed *ARF4* induction is a result of decreased post-transcriptional regulation by tasiARF.

Taken together, the induction of miR390 and *SGS3* would suggest a higher level of *TAS3* processing and eventually a higher level of tasiARF, but the case is the opposite: the tasiARF

production is compromised during virus infection, especially in the Bur ecotype. This observation suggests that the virus blocks one or more key steps of phased siRNA production. This can either be the miR390-mediated cleavage of the *TAS3* ncRNA or the RDR6/SGS3-mediated dsRNA precursor synthesis, or the phased cleavage of the dsRNA precursor by DCL4. The miR390-mediated cleavage is specific to the *TAS3* pathway, while the other components are shared by other tasiRNA-producing processes.

## Viral infection induces the production of tasiRNAs except for the tasiARF

In *Arabidopsis thaliana*, there are eight *TAS* loci of four families and many other phased siRNA-producing loci. Most of the sRNAs emerging from these loci are not AGO-competent and have no known biological role. However, some of them have evolved to target host genes. Only the *TAS3*-derived tasiARF is conserved among the land plants [24–26]. We wondered if the virus also blocks the production of other tasiRNAs or if this phenomenon is specific to the *TAS3* pathway, therefore, we analyzed their expression profiles. Surprisingly, unlike the tasiARF, all the investigated functional tasiRNAs were virus-induced (Fig 12), except for the phosphate starvation-induced TAS4-siR81 [39, 92] which is scarcely expressed under our circumstances.

These results suggest that the *TAS3* regulatory module is blocked at the *TAS3*-specific miR390 cleavage since all the other steps are carried out by shared components including the Bur virus-induced *SGS3* that might be responsible for the enhanced production of tasiRNAs from other loci. This blockage could happen either through ecotype-specific sequestering of miR390 or the AGO7 protein (or some unknown component of the tasiARF biogenesis

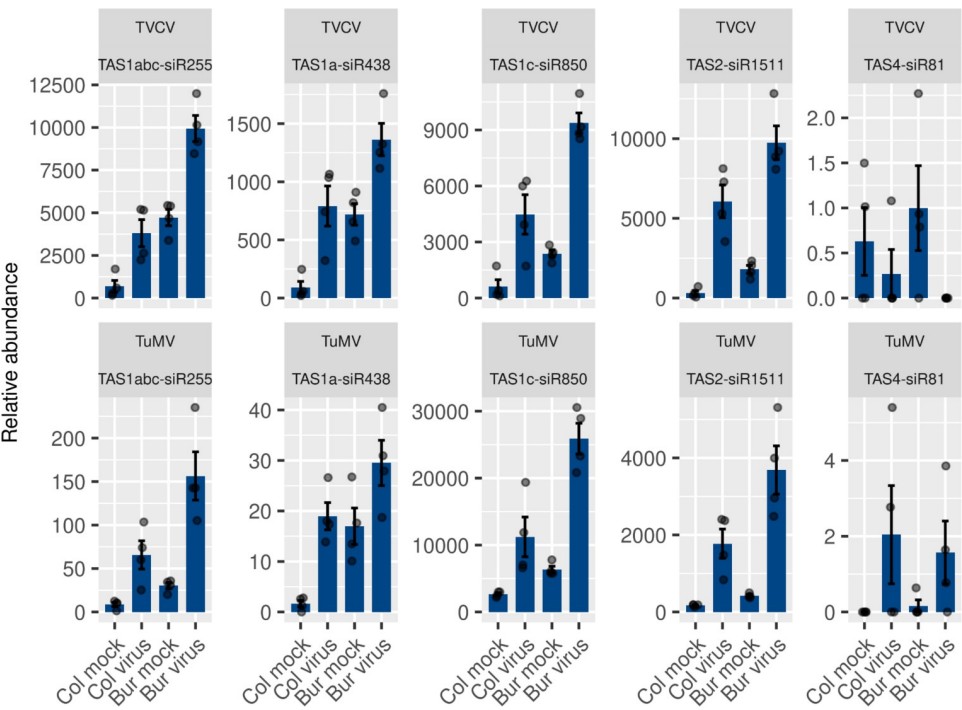

**Fig 12. Expression pattern of functional tasiRNAs other than the *TAS3*-derived tasiARF.** The normalized expression values of the functional siRNAs are shown in the TVCV and TuMV series. We considered only the mature, functional sequence; other sequence variants or non-functional siRNAs from the same TAS locus were not included in the analysis. Error bars represent the standard error of the mean of four biological replicates. The actual data points are also shown.

pathway) by the viral silencing suppressor. Alternatively, the AGO1-loading of tasiARF could be blocked which might lead to the destabilization of this siRNA specimen.

## QTL mapping suggests that the observed leaf deformation is a result of multigenic interaction

To support our findings, we performed a limited quantitative trait locus (QTL) mapping experiment using 43 selected recombinant inbred lines (RILs) between Bur and Col ecotypes that were established by INRA [77]. For the experiments, five plants per line were infected either with TVCV-ApH or mock and the leaf deformation was scored by inspection. The genotype data for the markers were obtained from INRA and along with the phenotype data, were used as input for the mapping software Windows QTL Cartographer [78]. A Single Marker Association test identified several markers on different chromosomes associated with the leaf deformation (S1 File). The strongest association was observed at the bottom of chromosome 5, where the *ARF4* (AT5G60450) is located (among others). However, the presence of other possible QTLs suggests that the leaf deformations are not caused by the erroneous expression of one gene (*ARF4*), rather, it might be the result of the interaction of multiple genes.

## Ecotype-specific genetic differences in the tasiARF biogenesis pathway components

We hypothesized that the VSRs could interfere with tasiARF biogenesis either by sequestering miR390 or by blocking some functions of AGO7, or some unknown component of the biogenesis pathway in an ecotype-specific manner. One possibility was that the structure of the miR390 sRNA duplex was different in the two ecotypes because of a possibly different sequence of the star strand, resulting in a differential binding by the VSRs. The sequence and structure of the miR390 sRNA duplex were shown to be important for selective binding by AGO7 [29, 93] and possibly affect its binding to VSRs as well. We investigated this possibility by predicting the miR390 loci (*MIR390A* on chromosome 2, and *MIR390B* on chromosome 5) in the Col and Bur ecotypes with ShortStack using the appropriate sRNA and genome sequences of the individual ecotypes separately. The analysis revealed no difference in the sequence of miR390 precursors or miRNA duplex structures.

Alternatively, VSRs could bind and block one of the components of the tasiARF biogenesis pathway in an ecotype-specific manner. To investigate this possibility, we compared the amino acid sequences of the proteins involved in the tasiARF biogenesis. We downloaded the protein sequences of AGO1, AGO2, AGO7, ARF4, and FIL from 19 different *Arabidopsis thaliana* ecotypes (including Bur and Col) from the homepage of the 1001 genome project (http://mtweb. cs.ucl.ac.uk/mus/www/19genomes/index.html), and aligned the sequence variants with Clustal omega [94]. According to the alignment, there were five amino acid changes in the Bur version of AGO7 protein compared to the Col version (all in the N-terminal part), while there were two and one amino acid changes in ARF4 and AGO2 proteins, respectively. There was no difference in the sequence of AGO1 and FIL proteins in the Col and Bur ecotypes (S2 Fig in S2 File). The Bur version of AGO7 was more different than the versions in the rest of the ecotypes. The two amino acid changes in ARF4 were not conserved (E/K and T/I in Col and Bur, respectively), and again, the Bur variant was rare among the investigated ecotypes. The FIL and AGO1 sequences were remarkably conserved among the investigated 19 ecotypes suggesting a strong selection pressure on them. There was some degree of variability in the AGO2 sequence but the Bur was not the most divergent ecotype in this respect.

Taken together, the best candidate for the ecotype-specific blockage of tasiARF production is AGO7 (or an unknown component of the TAS3 module) but the exact mechanism of blockage remains to be elucidated.

## Discussion

Viral symptom development is regulated by both viral and host factors. On the virus side, viral silencing suppressors were found to be the major symptom determinants. However, much less data is available for such symptom determinants on the plant side. During an infection, the virus modulates the molecular environment of the host cell in a way that favors the replication and spread of the virus. Since the viral genome has a limited coding capacity, the virus utilizes host factors for its replication, cell-to-cell movement, virion assembly, and release. Genetic variations in these host factors could affect the severity of viral symptoms. Such variations occur naturally and determine the host range in which the virus can replicate efficiently. Host factors can also be manipulated artificially which may result in a restricted viral replication or enhanced immune response [95, 96].

*Arabidopsis thaliana* is one of the most studied model organisms of plant biology. There are hundreds of ecotypes of this species that differ significantly in their geno- and phenotypes and therefore, are susceptible to viral infection differently. Indeed, in a genome-wide association study under field conditions, 317 *Arabidopsis* accessions were phenotyped for TuMV viral RNA accumulation and found significant differences among the susceptibility of the different ecotypes [97]. We also screened *Arabidopsis* ecotypes that display more severe symptoms upon virus infection (unpublished results) and we found that the Bur ecotype displays more severe leaf developmental defects than the Col ecotype when infected either with TVCV or TuMV. Since leaf development is regulated by sRNAs at many points [15, 16, 98, 99], we expected that the mRNA and sRNA profiles of the Col and Bur ecotypes would be significantly different when infected with viruses. Namely, we expected that genes and sRNAs that were responsible for the observed severe symptoms only in the Bur ecotype would change significantly (in the Bur virus sample only compared to the Col mock, Col virus, or Bur mock samples). To find such genes and sRNAs, we performed transcriptome and sRNAome analyses. Our analysis revealed that the levels of *FIL* (*YABBY1*, *AFO*) and *ARF4*, two transcription factors that are major abaxial polarity determinants in *Arabidopsis* leaf development [99], and the small RNA specimens tasiARF and vasiRNA-FIL that negatively regulates *ARF4* [30] and *FIL*, respectively, change significantly in the Bur ecotype when infected either with TVCV or TuMV, while such a change cannot be observed in the Col background. This led us to assume that the corruption of the *TAS3*-mediated regulatory module might be responsible for the observed leaf deformation in the virus-infected Bur plants. According to our analysis, miR390 is virus-induced in both ecotypes, especially upon TVCV infection but the tasiARF production is blocked only in the Bur virus sample. We observed that the production of other tasiRNAs is not blocked, rather, they are induced upon virus infection, especially in the Bur virus samples. This suggested that both viruses likely specifically block the *TAS3* pathway at the phase initiating AGO7-mediated cleavage. The Bur-specific depletion of tasiARF might lead to an ectopic expression of *ARF4* in the adaxial part of the leaf which could result in a misregulated auxin-dependent cell expansion in this domain, manifested as an upward curling leaf and wavy leaf margin.

ARF4 is called a repressive, class B ARF [100–102] because it lacks the glutamine-rich trans-activator domain that other activating ARFs have. Repressive ARFs might act by forming a dysfunctional heterodimer with other ARFs, and can form repressive chromatin in the promoter of the regulated gene [103]. In a previous study, *ARF8* was shown to be the major

symptom determinant in transgenic plants expressing three different viral silencing suppressors [104]. Unlike ARF4, ARF8 is an activating ARF and regulated post-transcriptionally by miR167 rather than tasiARF. In the *Arabidopsis* plants (Col background) overexpressing the TuMV silencing suppressor HCPro, *ARF4* was found to be upregulated that coincided with upward curling leaves, just like in our experiment. However, in the *arf4* mutant, the symptoms do not disappear, unlike in the *arf8* mutant. This might be because of the different background (the *arf4* mutant is in Col) that suggests a multigenic interaction that allows the symptoms to develop only in the Bur background.

In a different work that utilized a dexamethasone-inducible *FIL:FIL-GR* construct, researchers showed that FIL positively regulates *ARF4* and *KAN1* but not *ARF3* [85]. In our viral-infected samples, we observed the downregulation of *FIL* and upregulation of *ARF4* which seemingly contradicts the above-mentioned finding. However, the drop of the tasiARF level in our Bur virus sample in both the TVCV and TuMV series anti-correlates well with the *ARF4* level which might underline the importance of the post-transcriptional regulation of *ARF4*. It is also important to note that in the above-mentioned study, *FIL* was selectively induced in the abaxial part of the leaf where the *ARF4* is normally confined by the tasiARF gradient, while in our study, the *ARF4* expression was probably increased in the adaxial part due to the absence of inhibitory tasiARF. Also, the newly identified virus-induced vasiRNA-FIL might interfere with the delicate regulation of leaf polarity in an ecotype-specific manner.

Understanding the host-pathogen interaction network to diminish viral symptom development is one of the central goals of virological research. In this study, we have shown that two viruses having very different replication strategies, (TVCV and TuMV which are genuine representatives of *Tobamovirus* and *Potyvirus* families, respectively) cause very similar upward curling leaf and wavy leaf margin symptoms on the Bur but not on Col ecotype plants. The molecular basis for the ecotype difference therefore likely lies within the presence or absence of specific host factors that react in a very similar way to the infection by the two pathogens. By studying the transcriptomes and sRNAomes of the two sets of infections we hypothesize that the basis of the symptom development in Bur plants may be the enhanced impairment of the *TAS3* pathway. Both viruses possess a VSR that binds sRNAs, constituting a common target of action, for example, miR390 duplex binding by the two VSRs would impair *TAS3*-derived tasiARF biogenesis. An ecotype-specific structural difference in the miR390 duplex could result in selective binding by the VSRs. However, we could not find evidence for a different duplex structure of miR390a or miR390b in the two ecotypes. The RISC-loading efficiency of miR390 is very low and the overexpression of miR390 does not improve it, because the bottleneck is the availability of AGO7 [11]. It is possible that the miR390 loading into AGO7 is differentially affected by the presence of the virus either because the VSR directly manipulates the loading process or indirectly by relocating the AGO7 within the cell to a compartment that lacks miR390. Since some VSRs can bind ARGONAUTE proteins to block their antiviral action, we compared the amino acid sequences of the AGO proteins that could be involved in the production of tasiARF. There was no difference in the AGO1 sequence but AGO2 and especially AGO7 were more variable between the investigated ecotypes. We think that the best explanation is that the VSR binds and blocks some functions of the AGO7 protein selectively in the Bur ecotype but that would require a parallel evolution of the VSRs of different viruses to target the same AGO7 protein. Such parallel evolution of viral factors to target the same host factor is not unprecedented [105]. A study investigating the evolution of plant VSRs argues that plant VSRs are subject to episodic positive selection due to frequent jumps between different hosts rather than a one-to-one co-evolution with a single host [106]. This is consistent with our findings since both viruses we worked with have a broad host range (the turnip vein-clearing virus isolate we used, TVCV-ApH was isolated from *Alliaria petiolata*, a.k.a.

garlic mustard [45]). It is unclear why the virus would favor blocking AGO7, although it was described that AGO7 has some antiviral function [107, 108]. Another interesting possibility is that since the virus replication is often associated with an endomembrane system and the AGO7-mediated tasiRNA production is also endomembrane-associated [109], therefore, the viral replication could interfere with the AGO7-dependent tasiARF production. However, this interference should not be a passive process since it should happen only in the Bur ecotype.

Our QTL mapping results suggest that a multigenic interaction is behind the observed leaf deformations. Many of the genes that were selectively up- and downregulated in the Bur virus samples could be related to leaf development: many of the upregulated genes regulate cell wall structure while many downregulated genes are involved in auxin biosynthesis (i.e. *TAR2*) or signaling (i.e. *IAA29*), and could be directly involved in ARF4-mediated auxin signaling. It is worth mentioning that there are numerous transcription factors in both sets (see S3 Table) that could form an intricate regulatory network with or without ARF4 the net effect of which would lead to the observed phenotype. It is difficult to tell which players are the most upstream in the signaling chain that leads to the observed ecotype-specific leaf deformation, although we believe that the disrupted abaxial-adaxial gradient of tasiARF could be a major contributor. To reveal the exact nature of the selective blockage of tasiARF production in the virus-infected Bur ecotype, further research is needed in the future.

## Supporting information

**S1 Table. Sequencing statistics of the RNA-seq and sRNA-seq data.**
(XLSX)

**S2 Table. Normalized expression values of all *Arabidopsis* genes; Wald test results and normalized expression values of differentially expressed genes.**
(XLSX)

**S3 Table. Commonly regulated genes between the TVCV and TuMV series and their GO term enrichment analysis.** Content of the two sets (Bur virus up- and downregulated genes in both virus series), and the full result of their GO term enrichment analysis.
(XLSX)

**S4 Table. Wald test results and normalized expression values of differentially expressed sRNAs; commonly regulated sRNAs between the TVCV and TuMV series.**
(XLSX)

**S5 Table. Target prediction of the Bur virus upregulated sRNAs among the Bur virus downregulated genes.** Result of the target prediction performed using the psRNATarget server. Only the predicted Cleavage hits are shown. The sRNAs and the target genes are annotated.
(XLSX)

**S1 Data. Normalized expression values of all sRNAs in the samples.** Annotated expression table of sRNA sequences having a mean expression > 1. The table contains 102195 unique, 20–25-nt-long sequences ordered by mean expression value (highly expressed first).
(7Z)

**S1 File. Result of the QTL mapping.** The result of a single marker analysis performed by WinQTL Cartographer.
(PDF)

**S2 File.**
(PDF)

## Acknowledgments

We would like to thank Tibor Csorba for his valuable suggestions to improve the manuscript.

## Author Contributions

**Conceptualization:** György Szittya.

**Data curation:** Péter Gyula.

**Formal analysis:** Péter Gyula, Tünde Nyikó, Anita Sós-Hegedűs.

**Funding acquisition:** Péter Gyula, Tünde Nyikó, György Szittya.

**Investigation:** Péter Gyula, Tamás Tóth, Teréz Gorcsa, Tünde Nyikó, Anita Sós-Hegedűs.

**Methodology:** Péter Gyula, Tamás Tóth, György Szittya.

**Project administration:** Teréz Gorcsa, Tünde Nyikó, Anita Sós-Hegedűs, György Szittya.

**Resources:** György Szittya.

**Software:** Péter Gyula.

**Supervision:** György Szittya.

**Validation:** Tamás Tóth, Teréz Gorcsa, Anita Sós-Hegedűs.

**Visualization:** Péter Gyula.

**Writing – original draft:** Péter Gyula, Tamás Tóth.

**Writing – review & editing:** György Szittya.

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
