## [Decision Letter · Decision Letter 0]

25 Jul 2022

PONE-D-22-09203Ecotype-specific blockage of tasiARF production by two different RNA viruses in *Arabidopsis*PLOS ONE

Dear Dr. Gyula,

Thank you for submitting your manuscript to PLOS ONE. After careful consideration, we feel that it has merit but does not fully meet PLOS ONE’s publication criteria as it currently stands. Therefore, we invite you to submit a revised version of the manuscript that addresses the points raised during the review process.

We look forward to receiving your revised manuscript.

Kind regards,

Rajarshi Gaur

Academic Editor

PLOS ONE

Reviewers' comments:

Reviewer's Responses to Questions

**Comments to the Author**

1. Is the manuscript technically sound, and do the data support the conclusions?

Reviewer #1: Partly

Reviewer #2: Yes

2. Has the statistical analysis been performed appropriately and rigorously? 

Reviewer #1: Yes

Reviewer #2: Yes

3. Have the authors made all data underlying the findings in their manuscript fully available?

Reviewer #1: Yes

Reviewer #2: Yes

4. Is the manuscript presented in an intelligible fashion and written in standard English?

Reviewer #1: Yes

Reviewer #2: Yes

5. Review Comments to the Author

Reviewer #1: Comments to authors

Aims and methoda

Dr Gyula co-authors characterised the sRNA and RNA profiles of two Arabidopsis ecotypes in terms of interaction with two viruses belonging to different taxonomic groups. The aim of the study was to identify patterns/responses that could explain the common phenotype induced by the two viruses in the 'bur' ecotype alone and not in the 'col' ecotype.

Relevance of the study

The scientific questions posed by the authors are relevant, all the more so since plant ecotypes differenziate in precise geographic area and stabilise genetic traits whose study is a) in continuity with the establishment of platforms for the conservation and characterisation of plant genetic resources; b) fundamental for investigating and characterising traits of fast-changing environmental adaptation with regard to biotic and abiotic stresses.

The main results

The main answer in the study lies in the functionality of the long non-coding RNA called TAS3 and the abnormal production (abolished in the 'bur' ecotype) of tasiARFs. The results are consistent with the known and well-studied function of ARFs in leaf development (also well illustrated in the introduction and discussions). In itself, this finding might be sufficient to support the main conclusion.

In this frame, however, the authors discuss that AGOs involved in the function of tasiARFs could have differences in the two ecotypes, but they do not show it (different amino acid composition SNPs/indels at functional points) : the authors are invited to evaluate and show (if present) such differences available in the 1001 genomes project at https://1001genomes.org/.

Main criticisms

There are, however, some notable criticisms. The authors adopt a very interesting 'omic' approach to identifying genes involved in the symptomatological expression of the disease without definitely and fully considering (even only in the introduction) what has been discovered so far in terms of viral-associated siRNAs derived from coding genes. In particular, I refer to the work of Cao et al, 2012 (mentioned but not well discussed, being source of valuable information in the supplementary data concerning infection with TuMV, the same virus analysed in the present study), Leonetti et al, 2020 and Pitzalis et al, 2020. In particular, a cursory analysis of the Data_S1 table reveals that a large proportion of the unique sRNAs are 22-nt long (6,863, as opposed to 16,336 of 21-nt). This is why the authors, should take the opportunity of the knowledge they have gained (see leonetti et al., 2020 on the functionality of 22-nt vasiRNAs) and possibly extend it.

The authors are therefore invited to consider the issue of vasiRNAs as being responsible, together with tasiARFs. Alternatively they should eliminate the analysis of vasiRNAs from this study.

Moreover, with regard to vasiRNAs we note that there are no sRNAs in DATA_S1 that are derived from FIL and this finding is inconsistent with two of the panels in Figure 10 (i.e. vasiRNAs-FIL).

It is also unclear whether in the histograms of the figures, when referring to sRNAs from specific loci, only those of 21-nt are considered or also those of other lengths: to be clarified.

The authors are therefore invited to consider the issue of vasiRNAs as being responsible of the specific diseased phenotype, together with tasiARFs. Alternatively they should eliminate the analysis of vasiRNAs from this study.

Other criticisms

In the discussions an important concept is reported . The first part of the sentence is related to what is requested above (see “The main resuls” of comments to authors) but is not fully supported in the part concerning the evolution of the VSRs. In this regard, it is well known that VSRs of different viral taxonomic groups target different steps in the silencing pathways with different mechanisms, implying convergent evolution, i.e., having analogous features but non-homologous motifs. Strong VSR evolution/diversification is attributable to episodic selection rather than to pervasive positive selection (Murray et al., 2013). In other words, the diversification of plant virus VSRs is strongly boosted by frequent jumps between host species/genotypes rather than a one-to-one co-evolution. If what shown by Murray et al (2013) is in line with your discussion, please clarify.

Minor:

GWAS is only mentioned once in the discussions: do not use (unnecessary) acronym, which is instead required when the term is recurrent in the text more than 2-3 times.

Reviewer #2: Although this manuscript did not conclude with genetic examinations, this reviewer thinks it provides helpful suggestions for future studies to identify host factors involved in the symptom expressions the authors focused on, including leaf deformation caused by two distinct virus infections in Arabidopsis. The authors explored the host factors by comparison analysis with RNA-seq and small RNA-seq between virus-infected Col and Bur ecotypes. Bur showed leaf deformation with these virus infections, but Col did not. Their RNA-seq showed that two genes, ARF4 and FIL, were specifically induced and reduced in Bur. Then, sRNA-seq consistently revealed that tasiARF that regulates the ARF4 expression was specifically reduced among other tasiRNAs. QTL analysis with RILs between Bur and Col supported their transcriptome analyses; that is, one of the QTL detected was around the locus of ARF4, though the QTL analysis is perhaps considered preliminary level.

Minor points:

P2, L16, and L17, These ARF4 mean protein and thus should be roman.

P3, L17-22, These descriptions are of the case when plants have appropriate R genes that recognize invaded viruses. Rewrite them.

P25, L17, Fig 10 should be Fig 12

6. PLOS authors have the option to publish the peer review history of their article (what does this mean?). If published, this will include your full peer review and any attached files.

Reviewer #1: No

Reviewer #2: **Yes: **Yes

---

## [Author Response · Author response to Decision Letter 0]

6 Sep 2022

First of all, we appreciate the time and effort the reviewers invested to provide us with their valuable criticism and recommendations. We rewrote the introduction, results, and discussion part according to the suggestions which we believe significantly improved the manuscript. Please find our detailed reactions below.

Reviewer #1:

Aims and methods

Dr Gyula co-authors characterised the sRNA and RNA profiles of two Arabidopsis ecotypes in terms of interaction with two viruses belonging to different taxonomic groups. The aim of the study was to identify patterns/responses that could explain the common phenotype induced by the two viruses in the 'bur' ecotype alone and not in the 'col' ecotype.

Relevance of the study

The scientific questions posed by the authors are relevant, all the more so since plant ecotypes differenziate in precise geographic area and stabilise genetic traits whose study is a) in continuity with the establishment of platforms for the conservation and characterisation of plant genetic resources; b) fundamental for investigating and characterising traits of fast-changing environmental adaptation with regard to biotic and abiotic stresses.

The main results

The main answer in the study lies in the functionality of the long non-coding RNA called TAS3 and the abnormal production (abolished in the 'bur' ecotype) of tasiARFs. The results are consistent with the known and well-studied function of ARFs in leaf development (also well illustrated in the introduction and discussions). In itself, this finding might be sufficient to support the main conclusion.

In this frame, however, the authors discuss that AGOs involved in the function of tasiARFs could have differences in the two ecotypes, but they do not show it (different amino acid composition SNPs/indels at functional points): the authors are invited to evaluate and show (if present) such differences available in the 1001 genomes project at https://1001genomes.org/.

Answer: Thank you for your suggestion. We aligned the amino acid sequences of AGO1, AGO2, AGO7, ARF4, and FIL from 19 different ecotypes, including Col and Bur (Supplementary Fig S2). The analysis revealed that there were 5 amino acid changes in the Bur version of AGO7 compared to the Col version (all in the N-terminal part), while there were 0, 1, 2, and 0 amino acid changes in AGO1, AGO2, ARF4, and FIL, respectively (Fig S2). The Bur version of AGO7 was more different than the versions in the rest of the ecotypes. The 2 amino acid changes in ARF4 were not conserved (E/K and T/I in Col and Bur, respectively), and again, the Bur variant was rare among the investigated ecotypes. The FIL and AGO1 sequences were remarkably conserved among the investigated 19 ecotypes suggesting a strong selection pressure on them. There was some degree of variability in the AGO2 sequence but the Bur was not the most divergent ecotype in this respect.

Main criticisms

There are, however, some notable criticisms. The authors adopt a very interesting 'omic' approach to identifying genes involved in the symptomatological expression of the disease without definitely and fully considering (even only in the introduction) what has been discovered so far in terms of viral-associated siRNAs derived from coding genes. In particular, I refer to the work of Cao et al, 2012 (mentioned but not well discussed, being source of valuable information in the supplementary data concerning infection with TuMV, the same virus analysed in the present study), Leonetti et al, 2020 and Pitzalis et al, 2020. In particular, a cursory analysis of the Data_S1 table reveals that a large proportion of the unique sRNAs are 22-nt long (6,863, as opposed to 16,336 of 21-nt). This is why the authors, should take the opportunity of the knowledge they have gained (see leonetti et al., 2020 on the functionality of 22-nt vasiRNAs) and possibly extend it.

The authors are therefore invited to consider the issue of vasiRNAs as being responsible, together with tasiARFs. Alternatively they should eliminate the analysis of vasiRNAs from this study.

Answer: We admit that the discussion part could have been more thorough, especially regarding the research on the function vasiRNAs in plants (we fixed this in the Introduction and Discussion part, see the yellow text). It was because we focused on the differences rather than the similarities between the two ecotypes regarding their antiviral responses. We wanted to avoid discussing the commonly regulated sRNAs that were possibly not responsible for the differential symptoms observed in the two ecotypes. Additionally, we focused on the genes/sRNAs that were similarly differentially expressed only in the Bur ecotype infected by two different viruses and not discussed those genes/sRNAs that were unique for the two viruses or that were changed in the opposite direction. However, we observed that the vast majority of the commonly downregulated genes are potentially regulated by vasiRNAs that are commonly upregulated only in the Bur ecotype infected with both viruses. This is surprising, and therefore, we decided to discuss them in a little more detail, especially because these genes have functions that could be related to leaf development (i.e. auxin metabolism, cell wall expansion, etc).

Regarding the 22-nt-long vasiRNAs, the number of 22-nt sRNAs (6863) is the number of unique sequences in all the samples, including mock samples, therefore, it does not tell us how virus infection affects this size class. Actually, we found no such markedly increased population of this size class as in the case of CaMV infection investigated by Leonetti et al (see our corrected Fig 7). Maybe this is a CaMV-specific feature.

Moreover, with regard to vasiRNAs we note that there are no sRNAs in DATA_S1 that are derived from FIL and this finding is inconsistent with two of the panels in Figure 10 (i.e. vasiRNAs-FIL).

Answer: We are sorry for being confusing here: the vasiRNA-FIL is derived from a transposon (not from FIL) and potentially targets FIL in trans. It is analogous to tasi-ARF, when the functional siRNA is derived from the TAS3 non-coding RNA and targets ARF3- and ARF4-coding transcripts. We are aware that vasiRNAs could target the gene they derived from (in this case, a transposon). To predict potential vasiRNA/target pairs, we only considered those vasiRNAs and target genes that were differentially expressed only in the Bur virus samples in the opposite direction, i.e. upregulated vasiRNAs vs downregulated target genes. We did not predict all the possible targets of all the vasiRNAs, because we considered it irrelevant to our goal to find the possible cause of the differential leaf deformation.

It is also unclear whether in the histograms of the figures, when referring to sRNAs from specific loci, only those of 21-nt are considered or also those of other lengths: to be clarified.

Answer: When we show the expression of certain sRNA specimens (i.e. miR390a,b-5p, tasiARF a.k.a. TAS3 5’D7(+), TAS1abc-siR255, etc), we considered only the conserved, mature sequence; iso-miRs or other sequence variants (or other, non-functional siRNAs from the same TAS locus) were not included in the analysis.

Other criticisms

In the discussions an important concept is reported <>. The first part of the sentence is related to what is requested above (see “The main resuls” of comments to authors) but is not fully supported in the part concerning the evolution of the VSRs. In this regard, it is well known that VSRs of different viral taxonomic groups target different steps in the silencing pathways with different mechanisms, implying convergent evolution, i.e., having analogous features but non-homologous motifs. Strong VSR evolution/diversification is attributable to episodic selection rather than to pervasive positive selection (Murray et al., 2013). In other words, the diversification of plant virus VSRs is strongly boosted by frequent jumps between host species/genotypes rather than a one-to-one co-evolution. If what shown by Murray et al (2013) is in line with your discussion, please clarify.

Answer: We guess that the incriminated sentence was about the parallel evolution of the VSRs of different viruses targeting rice ARF17. We suppose that in our case the VSRs of the two different viruses interfere with the biosynthesis of the tasi-ARF in different ways. For example, one VSR might bind miR390 while the other might block AGO7 activity. As the reviewer pointed out, this would require non-homologous protein motifs. Both VSRs were reported to bind double-stranded sRNAs and HCPro was reported to bind AGO1. We analyzed the sequence of the miR390 duplex in the two ecotypes because we theorized that the different duplex structure (i.e. because of a possibly different star sequence) could result in a differential binding of miR390 duplex by the VSR(s). However, we found no evidence for such a difference. AGO1 amino acid sequences of the two ecotypes are also identical. We think that the best target of the VSRs is either the AGO7 or AGO2 proteins, or some unknown component of the tasi-ARF biosynthesis pathway. Murray et al argue that plant VSRs are subject to episodic positive selection due to frequent jumps between different hosts rather than a one-to-one co-evolution with a single host. This is consistent with our findings since both viruses we worked with have a broad host range (i.e. the turnip vein-clearing virus isolate we used, TVCV-ApH, was isolated from Alliaria petiolata, a.k.a. garlic mustard).

Minor:

GWAS is only mentioned once in the discussions: do not use (unnecessary) acronym, which is instead required when the term is recurrent in the text more than 2-3 times.

Answer: The Reviewer is right, we replaced the acronym with the full name.

Reviewer #2:

Although this manuscript did not conclude with genetic examinations, this reviewer thinks it provides helpful suggestions for future studies to identify host factors involved in the symptom expressions the authors focused on, including leaf deformation caused by two distinct virus infections in Arabidopsis. The authors explored the host factors by comparison analysis with RNA-seq and small RNA-seq between virus-infected Col and Bur ecotypes. Bur showed leaf deformation with these virus infections, but Col did not. Their RNA-seq showed that two genes, ARF4 and FIL, were specifically induced and reduced in Bur. Then, sRNA-seq consistently revealed that tasiARF that regulates the ARF4 expression was specifically reduced among other tasiRNAs. QTL analysis with RILs between Bur and Col supported their transcriptome analyses; that is, one of the QTL detected was around the locus of ARF4, though the QTL analysis is perhaps considered preliminary level.

Minor points:

P2, L16, and L17, These ARF4 mean protein and thus should be roman.

Answer: We changed the format to refer to protein products.

P3, L17-22, These descriptions are of the case when plants have appropriate R genes that recognize invaded viruses. Rewrite them.

Answer: Thank you for noticing this, we rephrased the sentence as suggested.

P25, L17, Fig 10 should be Fig 12

Answer: We fixed it.

---

## [Decision Letter · Decision Letter 1]

20 Sep 2022

Ecotype-specific blockage of tasiARF production by two different RNA viruses in *Arabidopsis*

PONE-D-22-09203R1

Dear Dr. Gyula,

We’re pleased to inform you that your manuscript has been judged scientifically suitable for publication and will be formally accepted for publication once it meets all outstanding technical requirements.

Kind regards,

Rajarshi Gaur

Academic Editor

PLOS ONE

**Comments to the Author**

1. If the authors have adequately addressed your comments raised in a previous round of review and you feel that this manuscript is now acceptable for publication, you may indicate that here to bypass the “Comments to the Author” section, enter your conflict of interest statement in the “Confidential to Editor” section, and submit your "Accept" recommendation.

Reviewer #1: All comments have been addressed

Reviewer #2: All comments have been addressed

2. Is the manuscript technically sound, and do the data support the conclusions?

Reviewer #1: Yes

Reviewer #2: Yes

3. Has the statistical analysis been performed appropriately and rigorously? 

Reviewer #1: Yes

Reviewer #2: N/A

4. Have the authors made all data underlying the findings in their manuscript fully available?

Reviewer #1: Yes

Reviewer #2: Yes

5. Is the manuscript presented in an intelligible fashion and written in standard English?

Reviewer #1: Yes

Reviewer #2: Yes

6. Review Comments to the Author

Reviewer #1: Thank you for accepting addressing all the and criticisms were raised during the first reviewing round.

Reviewer #2: (No Response)

7. PLOS authors have the option to publish the peer review history of their article (what does this mean?). If published, this will include your full peer review and any attached files.

Reviewer #1: No

Reviewer #2: No

---

## [Editor Report · Acceptance letter]

26 Sep 2022

PONE-D-22-09203R1 

Ecotype-specific blockage of tasiARF production by two different RNA viruses in *Arabidopsis*

Dear Dr. Gyula:

I'm pleased to inform you that your manuscript has been deemed suitable for publication in PLOS ONE. Congratulations! Your manuscript is now with our production department. 

Kind regards, 

on behalf of

Professor Rajarshi Gaur 

Academic Editor

PLOS ONE